# RETHINKING THE VULNERABILITY OF CONCEPT ERASURE AND A NEW METHOD

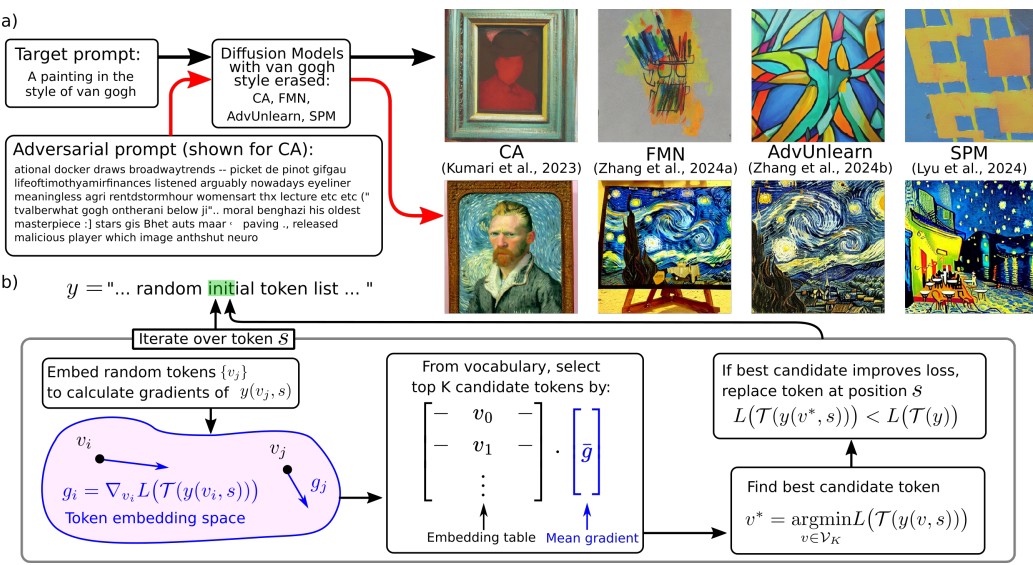

Figure 1: **a)** Examples images from models unlearned on van Gogh painting style. **b)** The update schematic of RECORD, which uses a linear gradient approximation to obtain a small set of candidate tokens, and then updates the prompt with respect to the exact evaluation of the loss function.

## ABSTRACT

The proliferation of text-to-image diffusion models has raised significant privacy and security concerns, particularly regarding the generation of copyrighted or harmful images. In response, concept erasure (defense) methods have been developed to "unlearn" specific concepts through post-hoc finetuning. However, recent concept restoration (attack) methods have demonstrated that these supposedly erased concepts can be recovered using adversarially crafted prompts, revealing a critical vulnerability in current defense mechanisms. In this work, we first investigate the fundamental sources of adversarial vulnerability and reveal that vulnerabilities are pervasive in the prompt embedding space of concept-erased models, a characteristic inherited from the original pre-unlearned model. Furthermore, we introduce **RECORD**, a novel coordinate-descent-based restoration algorithm that consistently outperforms existing restoration methods by up to 17.8 times. We conduct extensive experiments to assess its compute-performance tradeoff and propose acceleration strategies. The code for RECORD is available at [available upon publication].
Note: this paper may contain offensive or upsetting images

## 1 INTRODUCTION

Text-to-Image Diffusion models have recently garnered significant attention for their ability to generate high-quality images from natural language inputs (Song et al., 2020; Rombach et al., 2022). However, because these models are trained on vast and diverse datasets that may contain harmful

or undesirable content, their proliferation raises substantial ethical and safety concerns, particularly over the generation of copyrighted and harmful content (Chin et al., 2023; Somepalli et al., 2022). Pre-filtering undesired images from the training dataset is often considered impractical due to the sheer size of these datasets, as well as the cost of re-training models from scratch. Consequently, much research has pursued post-hoc approaches aiming to remove the undesired content from trained models via low-cost finetuning, while preserving the generation quality of other non-erased concepts (Gandikota et al., 2023a; Wu et al., 2024; Wu & Harandi, 2024; Zhang et al., 2024a; Kumari et al., 2023; Lyu et al., 2024; Fan et al., 2023; Gandikota et al., 2023b; Zhang et al., 2024b; Gong et al., 2024; Kim et al., 2024; Zhang et al., 2025; Gao et al., 2024; Srivatsan et al., 2025). This is commonly referred to as *concept erasure*, a subfield of *machine unlearning* (Kim & Qi, 2025).

However, it is well-known that neural networks are susceptible to adversarial attacks: small perturbations to an input can induce a well-trained model to produce any pre-determined outputs without altering the model itself (Kurakin et al., 2016; Dong et al., 2018; Yang et al., 2024; Beerens & Higham, 2024). This vulnerability raises similar concerns in the context of concept erasure. Indeed, recent studies have largely demonstrated the feasibility of eliciting unlearned models to re-generate the erased concepts via white-box optimization-based attacks (Chin et al., 2023; Zhang et al., 2023). We refer to this class of attack methods as *concept restoration*.

While the success of concept restoration methods demonstrates the vulnerability of concept-erased models, the underlying cause of this persistent susceptibility to adversarial attacks is not yet fully understood (Lu et al., 2025). To this end, our investigations reveal crucial insights into the fundamental vulnerability of unlearned models. Notably, prompt embeddings initialized in different regions of the embedding space, our findings suggest there often exist nearby embeddings that can restore the erased concepts. This suggests adversarial embeddings are pervasive in the prompt embedding space and can be exploited by the existing restoration algorithms. Furthermore, for certain types of erasure methods, embeddings initialized near the exact descriptions of the erased concept tend to diverge from those embeddings during optimization. This suggests that most existing unlearning methods only suppress the generation of the erased concept under prompts embedded near the specific prompt embeddings corresponding to the erased concept.

Stemming from these findings, we notice existing concept restoration methods rely on projecting the discrete text prompts into a continuous and differentiable space to enable gradient-based optimization (Chin et al., 2023; Zhang et al., 2023). However, recent studies have demonstrated that projection-based adversarial attacks generally underperform in comparison to coordinate-descend-based approaches (Carlini et al., 2023; Zou et al., 2023; Jones et al., 2023) in language model adversarial attacks. This motivates our investigation of similar approaches in the field of concept restoration. Therefore, we further propose **RECORD** (**R**estoring **E**rased **C**oncepts via Co**or**dinate **D**escent), a white-box coordinate-descent algorithm employing a two-stage optimization scheme to negate the need for projection (Figure 1). Our extensive experiments demonstrated that RECORD consistently achieve superior performance by up to 17.8-fold over the existing state-of-the-art restoration methods. Examples of the restored images are presented in Table 1.

The contributions of this paper are as follows:

- We explore why the majority of the current concept erasure methods are largely susceptible to concept restoration attacks.
- We extend the existing concept restoration attack methods by introducing RECORD, a coordinate descent approach motivated by similar successes on language model adversarial attacks.
- We conduct extensive ablation studies on RECORD, carefully assessing the effect of each hyperparameter and revealing its highly flexible compute-performance tradeoffs.

## 2 Background

### 2.1 Text-to-Image Diffusion Models

Diffusion Models are a class of generative model that generate images from text by learning to reverse the forward diffusion process. Starting with Gaussian noise $x_T \sim \mathcal{N}(0, \mathbf{1})$, a trained denoiser, commonly a U-Net (Ronneberger et al., 2015) or Vision Transformer (Dosovitskiy et al., 2020), iteratively denoises $x_T$ over the interval $t \in [0, T]$ until a clear image $x_0$ is reached. By conditioning

| Restoration Method | Erasure Method | | | | | |
|---|---|---|---|---|---|---|
| | ESD 2023a | FMN 2024a | AC 2023 | SPM 2024 | UCE 2023b | AdvUnlearn 2024b |
| P4D 2023 | | | | | | |
| UD 2023 | | | | | | |
| **RECORD** | | | | | | |

Table 1: Example images of erased concepts (van Gogh painting style) using token-level attacks. Each image column of the same concept is generated using the same latent initialization.

on prompt embeddings $c = \mathcal{T}(y)$, where $y$ is some natural language prompt, text-to-image generation is achieved. $\mathcal{T}$ is a pre-trained text encoder, commonly CLIP (Radford et al., 2021) or BLIP (Li et al., 2022). Latent diffusion models, such as Stable Diffusion (Rombach et al., 2022), perform the denoising in latent space $z_t = \mathcal{E}(x_t)$ and the denoiser $\epsilon_\theta$ is trained with the following objective

$$\underset{\theta}{\arg\min} \, \mathbb{E}_{z \sim \mathcal{E}(x), t, \epsilon \sim \mathcal{N}(0,1), c} \left\| \epsilon - \epsilon_\theta(z_t, t, c) \right\|_2^2 .$$

where $z_t$ is obtained from the forward diffusion process to the clean latent $z_0$ with Gaussian noise $\epsilon$.

## 2.2 PROMPT TUNING

Manipulating prompts to elicit specific behaviors from language models, also known as prompt tuning, is an important topic in Natural Language Processing research. (Ebrahimi et al., 2018) introduced HotFlip, generating adversarial examples through minimal character-level flips guided by gradients. Extending this, (Wallace et al., 2021) developed Universal Adversarial Triggers—input-agnostic token sequences optimized by using first order Taylor-expansion around the current token to select candidate tokens for exact evaluation. (Shin et al., 2020) presented AutoPrompt, designed to automatically generate prompts for various use cases. Addressing the lack of fluency in these prompts, (Shi et al., 2022) introduced FluentPrompt, incorporating fluency constraints and using Langevin Dynamics combined with Projected Stochastic Gradient Descent, where projection is done onto the set of token embeddings. (Wen et al., 2023) developed PEZ, an algorithm inspired by FluentPrompt, allowing for prompt creation in both text-to-text and text-to-image applications. In text-to-image models, (Gal et al., 2022) applied Textual Inversion, learning "pseudo-words" in the embedding space to capture specific visual concepts. Further advancements include GBDA (Guo et al., 2021), enabling gradient-based optimization over token distributions using the Gumbel-Softmax reparametrizaton (Jang et al., 2017) to stay on the probability simplex, GCG (Zou et al., 2023) and ARCA (Jones et al., 2023), optimizing discrete prompts via an improvement to AutoPrompt. ARCA will inspire our method.

## 2.3 CONCEPT RESTORATION

Recent methods for restoring erased concepts from unlearned models often leverage advanced optimization techniques similar to prompt tuning. Concept Inversion (CI) (Pham et al., 2023) introduces a new token with learnable embedding to represent the erased concept, which is learned to minimizes the reconstruction loss during denoising. This is a direct application of Textual Inversion (Gal et al., 2022) from prompt tuning to the concept restoration paradigm. Prompting Unlearned Diffusion Models (PUND) (Han et al., 2024) enhances this approach by iteratively erasing

and searching for the concept while also updating model parameters, improving transferability across models.

Other methods focus on discrete token optimization. UnlearnDiffAtk (UD) (Zhang et al., 2023) performs optimization over token distributions, similar to GBDA (Guo et al., 2021), but utilizes projection onto the probability simplex instead of the Gumbel-Softmax reparameterization. Prompting4Debugging (P4D) (Chin et al., 2023) optimizes prompts in the embedding space and projects onto discrete embeddings, akin to the approach used in PEZ (Wen et al., 2023). Additionally, Ring-A-Bell (Tsai et al., 2023) constructs an empirical representation of the erased concept by averaging embedding differences from prompts with and without the concept, then employs a genetic algorithm to optimize the prompt.

## 3 METHODS

### 3.1 MOTIVATION

Verifying whether a model has truly unlearned a concept is challenging. To assess the effectiveness of the unlearning process, we consider an unlearned denoiser $\epsilon_{\theta'}$ to be robust if it consistently fails to generate the erased content and produce images significantly different from those generated by the original model $\epsilon_\theta$ when subjected to any adversarial prompt and any latent initialization. Therefore, this work focuses on measuring the degree to which the unlearned model has diverged from the original model concerning the erased content. To achieve this, we propose a loss function similar to (Chin et al., 2023)

$$L(c) = \mathbb{E}_{t,z}\Big[ \left\| \epsilon_{\theta'}\big(z_t, t, c\big) - \epsilon_\theta\big(z_t, t, c_{\text{target}}\big) \right\|_2^2 \Big], \tag{1}$$

where $c = \mathcal{T}(y)$, $c_{\text{target}} = \mathcal{T}(y_{\text{target}})$, $z_t$ is obtained through the forward diffusion process with $z_0$ sampled from the target data distribution $p_{\text{target}}$. $y_{\text{target}}$ is the target prompt. The subsequent concept restoration attacks can be performed by minimizing this loss and finding the adversarial text prompt

$$y^* = \operatorname*{argmin}_y L(\mathcal{T}(y)).$$

This formulation is similarly applicable to erasure methods which unlearns the text encoder $\mathcal{T}$. This paper considers two types of restoration attacks to assess the vulnerability of unlearned models:

- **Embedding-level attacks**: In this setting, concept restoration is achieved by directly perturbing the prompt embedding $c$ to minimize the loss function defined in Equation [1]. With the prompt embedding space being continuous and differentiable, finding adversarial prompts poses an easier task. However, precise inversions from embeddings back to prompts are not guaranteed to exist, making embedding-level attacks less practicable and realistic in most circumstances.
- **Token-level attacks**: Directly perturbing prompt tokens to restore concepts is significantly more challenging due to their discrete and non-differentiable nature. To overcome this limitation, we introduce **RECORD** for carrying out robust concept restoration attacks.

### 3.2 EMBEDDING-LEVEL ATTACKS: A PEEK INTO THE ORIGIN OF MODEL VULNERABILITY

Embedding-level attacks for restoring the erased concept can be easily carried out in a naive approach of directly optimizing on the prompt embedding $c$ with a fixed learning rate. Since computing the exact expectation over all latents and timesteps is intractable, we approximate the $L(y)$ from Equation [1] as:

$$\hat{L}(c, \mathcal{Z}) = \sum_{(z_t, t) \in \mathcal{Z}} \left\| \epsilon_{\theta'}\left(z_t, t, c\right) - \epsilon_\theta\left(z_t, t, \mathcal{T}\left(y_{\text{target}}\right)\right) \right\|_2^2, \tag{2}$$

where $\mathcal{Z}$ is a sampled batch of noised images and their corresponding timesteps. For embedding-level attacks, we use a batch size of 16 images and NAdam for optimizing onprompt embedding $c$ with a fixed learning rate of 0.1. The full embedding-level attack results can be found in Appendix B.

To explore vulnerability, we visualize the 2D isomap (Wang, 2012) projections of prompt embedding optimization trajectories (Figure 2a), providing a visual intuition for their flow within the embedding space. We use an exact description of the erased concept as the reference target, e.g. "a painting in the style of van Gogh" for models unlearned on van Gogh art style, and investigate four different

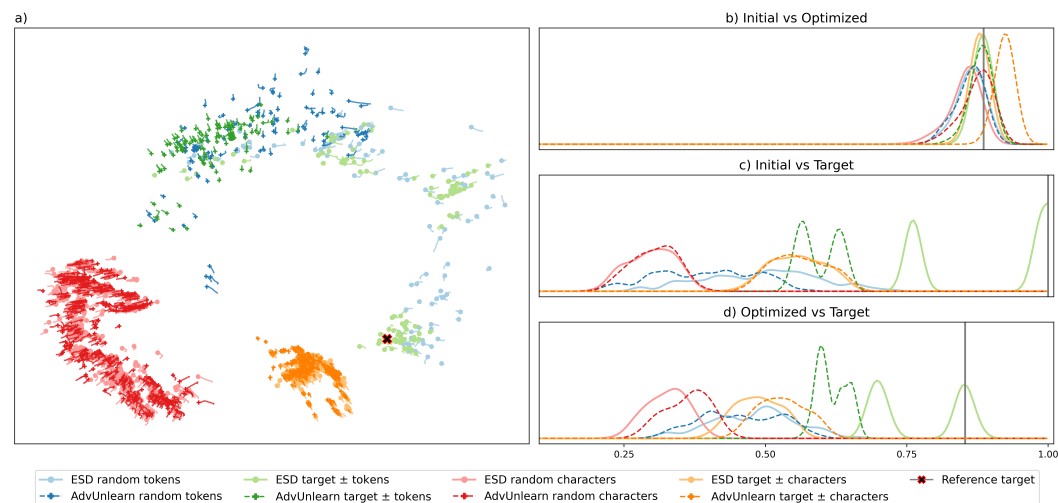

Figure 2: Behavior of the text embeddings during embedding-level attacks on models unlearned with ESD and AdvUnlearn. **a)** Isomap projection of the optimization trajectories in the prompt embedding space $\mathbb{R}^{T \times 77 \times 768}$ down to $\mathbb{R}^{T \times 2}$. 2000 trajectories shown, each $T = 10$ steps long. Dots / crosses denote the starting point. The erased concept can be generated at the end of each trajectory. **b)**, **c)**, **d)** present the cosine similarity histogram, computed in $\mathbb{R}^{77 \times 768}$, between the initial, optimized, and reference target embeddings.

initialization schemes: Prompt embeddings initialized "close" to the reference target of the erased concept by padding and replacing random tokens ($\pm$ tokens) or characters ($\pm$ characters) of random lengths; Prompt embeddings initialized "far" from the reference target by uniformly sampling random tokens or characters. Remarkably, embeddings initialized in all four regions of the embedding space can successfully restore the erased concept with short optimization trajectories (Figure 2a and 2b). In particular, embeddings initialized in the far region can generate the erased concept without approaching the region close to the reference target embedding. This suggests that, there often locally exists an adversarial prompt embedding close to the initialization. In other words, unlearned models are generally vulnerable to small perturbations to the text embedding, and effective adversarial embeddings are widespread in the prompt embedding space. This echoes the well-established understanding that neural networks are adversarially vulnerable to small perturbation to its inputs (Fawzi et al., 2018; Simon-Gabriel et al., 2018; Wu et al., 2020; Beerens & Higham, 2024).

For denoiser-based erasure methods like ESD (Gandikota et al., 2023a), embeddings initialized "close" to the target embedding tend to diverge from the reference target embedding during optimization (Figure 2c and 2d). This suggests that denoiser-based erasure methods only suppress generation of a concept in a localized region around the reference target embedding. By contrast, for methods that unlearn the text encoders, such as AdvUnlearn (Zhang et al., 2024b), embeddings tend to converge slightly towards yet still remain far away from the reference target. This dynamic indicates a different failure mode: modifying the text encoder alters the token-embedding mapping but does not necessarily erase the model's inherent ability to generate the concept from embeddings located in that specific region.

Collectively, these results demonstrate that embedding-level vulnerabilities are ubiquitous and remain largely unaddressed except in some specific regions of the embedding space. The pervasive nature of these embeddings make the concept-erased models susceptible to exploitation by token-level restoration algorithms. Conversely, for erasure methods that are more robust to embedding-level attacks, such as SH (Wu & Harandi, 2024) (see Appendix B), token-level attacks are also likely to underperform. Building on this, we further elaborate in Appendix A that the pervasiveness of these vulnerabilities is not an artifact of the concept erasure process, but inherited from the original pre-unlearned model: *it is possible to find local embeddings for generating a target concept even when initializing from a semantically distant starting point.*

**Algorithm 1** Pseudocode of RECORD.

```
# θ': original model; θ: unlearned model
# y_target: target prompt
# J: gradient samples; K: candidates
# S: sequence length; N: passes
# R: reference set;  E: embedding table

Random token sequence y of length S: # initialization
for n=1 to N: # load N passes
    # Shuffle
    Random permutation π of positions {1,···,S}

    for s in π:
      # sample data
      Sample batch Z of noise images and timesteps
      # candidate selection
            │  Sample J random tokens {v_j}
            │  Compute gradients ḡ = 1/J Σ_{j=1}^{J} ∇v_j L̂(T(y(v_j,s)), Z(n,s))
            │  Score all tokens:scores ← E ḡ
            │  Select top K tokens V based on scores
      # candidate evaluation
      v̂* = arg min_{v∈V} L̂(y(v,s),  Z) # best candidate
      # coordinate descent
      if L̂(T(y(v̂*,s)),  R) < L̂(T(y),  R):
          Update y ← y(v̂*,s)
Return:  optimized prompt y for restoring erased concepts
```

## 3.3 TOKEN-LEVEL ATTACKS: NEW METHOD

Existing concept restoration methods use gradient-based optimization, which necessitates the projection of the non-differentiable discrete text prompts to a continuous and differentiable space (Chin et al., 2023; Zhang et al., 2023). However, recent studies have observed superior performance of coordinate-descend-based methods over their projection counterparts when optimizing on discrete texts (Carlini et al., 2023; Zou et al., 2023; Jones et al., 2023). This motivates us to introduce RECORD, a coordinate descent approach that iteratively optimizes the prompt by refining one token at a time while fixing all other tokens. A naive implementation of this strategy requires evaluating the loss function for every token in the vocabulary at each position, which quickly becomes intractable for large vocabularies. To make this optimization feasible, RECORD uses a two-stage approach of leveraging a linear approximation of the loss gradient to identify a small subset of candidate tokens, then perform exact evaluations to determine the optimal token for substitution.

More precisely, the algorithm first initializes a random token sequence $y = [y_1, \ldots, y_S]$ of length $S$, and iteratively performs $N$ passes over $y$. In each pass, a random permutation of the token positions is generated to mitigate positional bias during updates. For each position $s$ in the permuted sequence, the algorithm samples a batch of clean latents $z_0^{[n]}$ and corresponding timesteps $t^{[n,s]}$. Candidate tokens $v$ for position $s$ are then selected by sampling $J$ random tokens, and computing the gradient $g$ of the loss $\hat{L}$ from Equation [2] with respect to the candidate token embeddings:

$$g_j = \nabla v_j \hat{L}(\mathcal{T}(y(v_j, s)), \mathcal{Z}(n, s)).$$

The average gradient $\bar{g} = \frac{1}{J} \sum_{j=1}^{J} g_j$ serves as a linear approximation to $\hat{L}$ with respect to the entire prompt embedding space. By multiplying the embedding table $E$ with $\bar{g}$, we efficiently score all possible tokens and select the top $K$ candidates for exact evaluation. This effectively alleviates the intractability introduced by the large vocabulary in the naive approach. During the evaluation and subsequent update of $y_s$, we employ a greedy strategy: a candidate token $\hat{v}^*$ is only accepted if it improves the loss, i.e. when $\hat{L}(\mathcal{T}(y(\hat{v}^*, s)), \mathcal{R}) < \hat{L}(\mathcal{T}(y), \mathcal{R})$, where $\mathcal{R}$ is the reference set. This update process iterates through all positions in the permutation and repeats for $N$ passes, progressively enhancing the token sequence over time. Since each accepted token replacement strictly decreases the loss and the number of possible token sequences is finite, the algorithm is guaranteed to converge to a coordinate-wise local minimum. A pseudocode can be found in Algorithm 1.

Although the RECORD algorithm as described above is tailored for attacking denoiser-based erasure methods, it can be easily adapted to text-encoder-based erasure methods by replacing $\mathcal{T}$ with $\mathcal{T}'$ in $\hat{L}$ when encoding $y$, where $\mathcal{T}'$ is the unlearned text encoder.

| Erased concept | van Gogh | Church | Garbage Truck | Parachute | Nudity |
|---|---|---|---|---|---|
| **Accuracy** | 99.4 | 98.8 | 93.4 | 84.0 | 87.6 |

Table 2: The classification accuracy of Stable Diffusion 2.1 as a zero-shot image classifier

## 4 EXPERIMENTS

We designed our experiments to address the following questions:

1. Does the proposed RECORD algorithm outperform current SOTA concept restoration methods?

2. Are certain concept erasure methods more robust?

3. How important are the different RECORD hyperparameters?

We extensively compare RECORD against the two current state-of-the-art concept restoration methods in the literature, P4D (Chin et al., 2023) and UD (Zhang et al., 2023), on text-to-image diffusion models unlearned with both denoiser-based (ESD (Gandikota et al., 2023a), ED (Wu et al., 2024), SH (Wu & Harandi, 2024), FMN (Zhang et al., 2024a), CA (Kumari et al., 2023), SPM (Lyu et al., 2024), SalUn (Fan et al., 2023), UCE (Gandikota et al., 2023b), and RECE (Gong et al., 2024)) and text-encoder-based (AdvUnlearn (Zhang et al., 2024b)) concept erasure methods. Although there are other black-box methods that suppress generation of harmful content without post-hoc training the denoiser or the text encoder (Yoon et al., 2024; Schramowski et al., 2023; Li et al., 2024; Jain et al., 2024), such methods target a fundamentally different adversary, e.g. one with only API access, and cannot exploit internal gradients or weights of the unlearned model. Including them here would not yield an apple-to-apple evaluation of concept erasure under full model access. We use open-sourced unlearned model weights from (Zhang et al., 2024b; Gong et al., 2024), which uses Stable Diffusion 1.4 (SD1.4) (Rombach et al., 2022) as the base model. The erased concepts include art style (van Gogh), objects (church, garbage truck, parachute) and nudity.

### 4.1 EVALUATION METRIC

Most text-to-image diffusion models can generate a far broader range of objects and styles than any single image classifier is capable of classifying. Prior work therefore evaluates concept erasure and restoration methods by ensembling multiple classifiers, each with its own architecture, training data, and preprocessing method, introducing potential inconsistencies. To address this issue, as well as to improve reproducibility and ease replication, we adopt a single, unified zero-shot diffusion classifier (Stable Diffusion v2.1) (Li et al., 2023; Clark & Jaini, 2023). The classification results are obtained by computing

$$y^* = \underset{y_i \in Y}{\arg\min} \, \mathbb{E}_t \, \|\epsilon - \epsilon_\theta(z_t, t, \mathcal{T}(y_i))\|_2^2 ,$$

where the timestep $t$ is uniformly sampled from $U(0, T)$, $Y = \{y_1, y_2, \ldots, y_n\}$ is a set of $n$ classification classes. The expectation is computed over 10 samples, which in our experience is sufficient to provide accurate classification results. For art style and object attacks, we build sets of 50 classes using prompt templates 'a painting in the style of {artist_name}' and 'a photorealistic image of {object}', where the artist names are randomly chosen from a list of famous painters, e.g. Leonardo Da Vinci, and object names are from the classification classes of YOLOv3 (Redmon & Farhadi, 2018). For nudity attacks, a set of 4 classes are built with the same template as the object attacks. For all attacks, we also additionally add one empty class, ' ', which helps the classifier capture images that fall significantly outside the distributions of the specified classes. All results presented in this section are computed on 500 images generated by the corresponding erased models and restoration attacks. We report the Attack Success Rate (ASR) in percentage, calculated by dividing the number of images classified as the target (erased) class by the total number of generated images. We attach the classification accuracy of the zero-shot diffusion classifier on images generated by the baseline model in Table 2, which serves as references for the ASRs of the corresponding concepts.

| Erased Concept | Restoration Method | Erasure Method | | | | | | |
|---|---|---|---|---|---|---|---|---|
| | | ESD 2023a | FMN 2024a | AC 2023 | SPM 2024 | UCE 2023b | AdvUnlearn 2024b | RECE 2024 |
| van Gogh | P4D 2023 | 6.6 | 27.2 | 49.8 | 54.8 | 67.2 | 2.8 | 50.8 |
| | UD 2023 | 5.4 | 25.4 | 17.0 | 34.6 | 42.8 | 2.8 | 10.0 |
| | **RECORD** | 64.0 | 76.8 | 94.0 | 95.6 | 97.6 | 33.0 | 89.0 |

| Erased Concept | Restoration Method | Erasure Method | | | | | | |
|---|---|---|---|---|---|---|---|---|
| | | ED 2024 | ESD 2023a | SalUn 2023 | SH 2024 | SPM 2024 | AdvUnlearn 2024b | RECE 2024 |
| Church | P4D 2023 | 16.0 | 24.6 | 28.8 | 3.4 | 51.6 | 7.0 | 8.2 |
| | UD 2023 | 2.6 | 4.8 | 5.4 | 4.4 | 22.8 | 1.4 | 6.8 |
| | **RECORD** | 61.2 | 75.2 | 71.4 | 8.6 | 92.2 | 57.0 | 46.4 |
| Garbage Truck | P4D 2023 | 9.4 | 18.8 | 21.0 | 0.4 | 35.8 | 34.2 | 5.6 |
| | UD 2023 | 16.0 | 3.8 | 17.0 | 4.4 | 29.2 | 0.2 | 1.2 |
| | **RECORD** | 40.8 | 38.8 | 58.0 | 1.0 | 66.4 | 50.0 | 17.0 |
| Parachute | P4D 2023 | 5.6 | 11.6 | 20.6 | 0.6 | 15.6 | 2.0 | 4.2 |
| | UD 2023 | 3.0 | 2.4 | 2.4 | 1.0 | 6.8 | 1.2 | 3.2 |
| | **RECORD** | 15.4 | 44.6 | 48.8 | 2.0 | 60.4 | 35.6 | 10.0 |
| Nudity | P4D 2023 | 2.0 | 37.6 | 9.8 | 9.6 | 28.8 | 17.0 | 15.2 |
| | UD 2023 | 4.0 | 19.2 | 4.2 | 2.0 | 2.5 | 14.2 | 19.4 |
| | **RECORD** | 2.4 | 70.6 | 9.0 | 21.2 | 69.0 | 39.2 | 38.8 |

Table 3: Attack success rate (%) for models erased on different concepts (van Gogh style, Church, Garbage Truck, Parachute, Nudity), compared with different restoration methods P4D (Chin et al., 2023), UD (Zhang et al., 2023), and RECORD. The best and second-best values are marked in red and gray, respectively.

## 4.2 RESULTS

In this experiment, each restoration method (P4D (Chin et al., 2023), UD (Zhang et al., 2023), RECORD) aims to find 64-token-long seed-agnostic adversarial prompts starting from a randomly initialized prompt sequence, except UD. Since UD optimizes on a token distribution, we follow their original initialization strategy by setting the first few tokens to be the target prompt, and initializing the rest of the tokens from a uniform distribution for all tokens. Without this type of initialization UD does not achieve any significant results. Each restoration method is evaluated by identifying 50 adversarial prompts on an H100 GPU and using which to generate 500 images per method for ASR calculations. For RECORD, we use $N = 20$ passes through the token list, a batch size of 1 image each, and $J = 64$ samples for the candidate selection. The chosen candidate set has size $K = 64$. Example images can be found in Table 1 and Appendix K.

RECORD consistently outperforms P4D and UD (Table 3) by up to 17.8 times (see the AdvUnlearn-Parachute cells), except for a few minor exceptions. In particular, AdvUnlearn is quite resilient against P4D and UD with single digit ASR on most concepts, while RECORD is able to achieve an ASR of at least 33% for all concepts. Additionally, different erasure methods seem to have different level of robustness against adversarial attacks on different concepts. For example, ED (Wu et al., 2024) is more robust in erasing nudity-related concepts, but not on objects such as church. Similar observation can also be observed in Appendix F. On the other hand, SH models are very robust against all concept restoration attacks, but this comes at a significant cost of the generation quality of other non-erased concepts, which has been discussed in detail by (Zhang et al., 2024b). We additionally conduct fixed-seed ablation study and demonstrate RECORD still perform well in Appendix E.

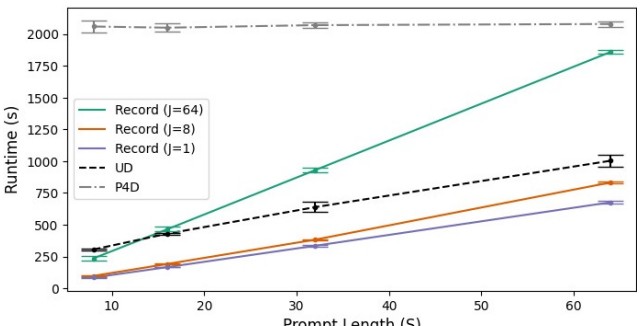

Figure 3: The mean runtime and the standard deviation (annotated as error bars) of different restoration methods, computed over 10 runs and at sequence lengths $S = 8, 16, 32, 64$. We note it is possible to achieve substantial acceleration by lowering gradient token number $J$, with only marginal performance loss, as discussed in Appendix D.

## 5 DISCUSSIONS

To validate the design of RECORD, we conducted several ablation studies. First, we addressed the ongoing debate in the literature regarding the optimal loss function for concept restoration (Pham et al., 2023; Zhang et al., 2023; Chin et al., 2023). Our findings in Appendix B show that the loss function in Equation [1], which uses the original model's noise prediction as a target, marginally outperforms alternatives that rely on Gaussian noise. This is because the original model's predictions act as a more informative surrogate, justifying its slight increase in computational overhead.

The runtime of RECORD is very competitive with existing restoration methods (Figure 3), and its hyperparameters $S, N, J, K$ provide flexible control over the compute-performance trade-off (Appendices C,D). For example, our ablation studies in Appendix D demonstrate that a significant 60% acceleration can be achieved by lowering the number of gradient tokens $J$ with only a marginal loss in performance. This allows RECORD's runtime to surpass that of P4D and UD while maintaining its superior ASRs. Additionally, our studies on prompt length $S$ and the number of passes $N$ conclude that the best strategy for maximizing ASR for a given compute budget is to choose higher $S$ and lower $N$, as the breath of the search space is more significant than the depth. These results can be found in Appendix C.

We also tested the transferability of successful adversarial prompts from SD1.4 to larger unlearned models like SDXL (Podell et al., 2024) and FLUX (Labs et al., 2025) in Appendix F. Our results show that adversarial prompts identified by RECORD are generally more transferable than those from P4D and UD. This finding has significant black-box implications, as an adversary could use a prompt optimized on an open-sourced model to attack a different black box model without requiring excess to the model weights. This superior level of transferability allows RECORD to mitigate the limitations of the white-box assumption commonly used by the existing concept restoration methods.

Lastly, we investigate the scalability of RECORD on a more challenging setting i.e. restoring erased concepts on more sophisticated text-to-image models, such as SDXL and FLUX. In particular, these larger models have a dual-encoder setup for encoding prompts, which differs from the single text encoder used in SD1.4. This introduces additional difficulties in restoration algorithm design and is rarely addressed in the existing concept restoration literature (Zhang et al., 2025). We assess five different strategies for adapting RECORD to handle this architectural difference and provide some initial results (Appendix G). Example images by SDXL and FLUX are showcased in Appendix L.

## 6 CONCLUSIONS

In this study, our investigation into existing concept erasure methods used in text-to-image diffusion models reveal that adversarial prompt embeddings are pervasive throughout the embedding space, which can be exploited by token-level concept restoration methods. We further introduce RECORD, a novel token-level concept restoration algorithm designed for restoring erased concepts by adversarially perturbing the input prompts in a coordinate-descent manner. We conduct extensive experiments and ablation studies demonstrating the consistent superiority of RECORD as well as the effect of its hyperparameters. These results not only underscore significant vulnerability inherent in current erasure approaches, but also pave the way for the future development of erasure and restoration algorithms that can more effectively mitigate or exploit these vulnerabilities.

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

# A "VULNERABILITY" ANALYSIS ON THE ORIGINAL MODEL

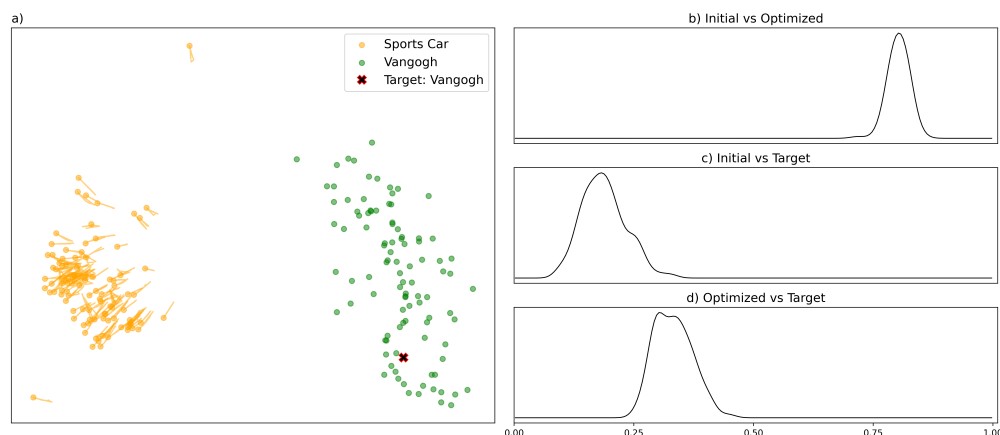

Figure 4: Behavior of the text embeddings during embedding-level attacks on original model. **a)** Isomap projection of the optimization trajectories of the prompt embeddings. Dots denote the starting point of the trajectory. The target concept can be generated at the end of each trajectory. **b)**, **c)**, **d)** are smoothened cosine similarity histograms between the initial, optimized, and target embeddings.

Extending the discussion from Section 3.2, which established the pervasive existence of adversarial embeddings in unlearned models, this section investigates the fundamental origin of this vulnerability. We seek to answer a critical question: are these vulnerabilities an artifact of the concept erasure process, or are they an intrinsic property of the original, pre-unlearned diffusion model?

To explore this, we conduct an experiment on the original, pre-unlearned model using the same embedding-level optimization methodology described in Section 3.2. The objective is to determine if prompt embeddings capable of generating a specific target concept could be discovered starting from a completely irrelevant and semantically distant initial concept.

We use "a painting in the style of van Gogh" as the target prompt, and "a photorealistic image of a sports car" as the semantically distant starting point. We generate 100 paraphrases of the starting prompt and target prompt using Gemini 2.5 and encode them as the embedding initializations and as the reference cluster, respectively.

Our findings reveal that even when initialized with embeddings for a "sports car," the optimization process consistently discovers local embeddings that can generate van Gogh style paintings, far from the region associated with the actual "van Gogh" prompt (Figure 4).

This result provides a crucial insight: the high-dimensional embedding space of the original model is densely populated with regions that can trigger the generation of a given concept. The vulnerability to concept restoration is therefore not primarily induced by the unlearning process but is an characteristic inherited from the pre-unlearned model. Most concept erasure techniques focus on suppressing a concept in the localized region around its explicit text description, leaving these numerous, distant regions of vulnerabilities adversarially exploitable by the restoration attacks analyzed in this paper.

## B  EMBEDDING-LEVEL ATTACK RESULTS AND LOSS FUNCTION COMPARISON

| Erased Concept | Restoration Method | Erasure Method | | | | | |
| --- | --- | --- | --- | --- | --- | --- | --- |
| | | ESD 2023a | FMN 2024a | AC 2023 | SPM 2024 | UCE 2023b | AdvUnlearn 2024b |
| van Gogh | No attack | 3.4 | 17.6 | 32.2 | 45.4 | 52.8 | 0.8 |
| | Embed attack 1 | **99.8** | **99.8** | 99.4 | **100.0** | 91.4 | 92.8 |
| | Embed attack 2 | **99.8** | **99.8** | **100.0** | 99.4 | **98.4** | **99.0** |

| Erased Concept | Restoration Method | Erasure Method | | | | | |
| --- | --- | --- | --- | --- | --- | --- | --- |
| | | ESD 2023a | ED 2024 | SH 2024 | SPM 2024 | SalUn 2023 | AdvUnlearn 2024b |
| Church | No attack | 20.2 | 4.4 | 1.0 | 84.6 | 2.0 | 1.6 |
| | Embed attack 1 | 95.6 | 99.4 | 22.8 | 98.0 | 99.4 | 96.6 |
| | Embed attack 2 | **100.0** | **100.0** | **64.6** | **99.8** | **100.0** | **100.0** |
| Garbage Truck | No attack | 6.2 | 20.4 | 9.2 | 28.4 | 9.0 | 0.2 |
| | Embed attack 1 | 98.0 | 98.4 | **18.0** | 98.0 | 99.6 | **96.2** |
| | Embed attack 2 | **98.8** | **100.0** | 3.8 | **99.6** | **100.0** | 92.4 |
| Parachute | No attack | 0.8 | 3.8 | 2.2 | 29.8 | 1.2 | 0.2 |
| | Embed attack 1 | 91.2 | 89.6 | **4.8** | **73.6** | **63.0** | **46.6** |
| | Embed attack 2 | **99.4** | **93.8** | 1.2 | 46.6 | 61.6 | 44.8 |

| Erased Concept | Restoration Method | Erasure Method | | | | | |
| --- | --- | --- | --- | --- | --- | --- | --- |
| | | ESD 2023a | SH 2024 | SPM 2024 | SalUn 2023 | UCE 2023b | AdvUnlearn 2024b |
| Nudity | No attack | 49.2 | 2.0 | 49.6 | 1.8 | 41.8 | 38.4 |
| | Embed attack 1 | **98.4** | 83.0 | **99.6** | 99.4 | 96.8 | **98.6** |
| | Embed attack 2 | 98.2 | **87.6** | 98.6 | **99.8** | **98.8** | 96.8 |

Table 4: Attack success rate (%) of the embedding-level attacks for art style, object and nudity attacks. Embed attack 1 and 2 refer to the two loss formulation $\widetilde{L}_1$ and $\widetilde{L}_2$, respectively.

The selection of an optimal loss function for identifying adversarial prompts to restore concepts from unlearned models is a topic of active debate (Pham et al., 2023; Zhang et al., 2023; Chin et al., 2023). This ablation study seeks to resolve this ambiguity by empirically comparing two prominent loss functions:

$$L_1(y) = \mathbb{E}_{t,z}\left[\left\|\epsilon_{\theta'}\left(z_t, t, \mathcal{T}(y)\right) - \epsilon\right\|_2^2\right] \tag{3}$$

$$L_2(y) = \mathbb{E}_{t,z}\left[\left\|\epsilon_{\theta'}\left(z_t, t, \mathcal{T}(y)\right) - \epsilon_\theta\left(z_t, t, \mathcal{T}(y_{\text{target}})\right)\right\|_2^2\right] \tag{4}$$

where $z_t$ is obtained through the forward diffusion process, $z_0 \sim p_{\text{target}}$ is sampled from the target data distribution, $y_{\text{target}}$ is the target prompt. $L_1$ (Pham et al., 2023) and $L_2$ (Chin et al., 2023) are minimized by optimizing prompts $y$ to match the denoiser predictions respectively to: the true noise from the forward diffusion sequence $\epsilon$, or the predicted noise by the baseline denoiser with the target prompt $\epsilon_\theta\left(z_t, t, \mathcal{T}(y_{\text{target}})\right)$.

To compare both loss functions, we consider the embedding-level attack setting:

$$\widetilde{L}_1(c) = \mathbb{E}_{t,z}\left\|\epsilon_{\theta'}(z_t, t, c) - \epsilon\right\|_2^2 \tag{5}$$

$$\widetilde{L}_2(c) = \mathbb{E}_{t,z}\left\|\epsilon_{\theta'}(z_t, t, c) - \epsilon_\theta(z_t, t, c_{\text{target}})\right\|_2^2. \tag{6}$$

We use NAdam optimizer and iterate over a sampled $z_0$ set of 100 images 50 times, with a constant learning rate of 0.1 and a batch size of 16. $z_0$ is generated by the baseline model using the target

prompt. Attacks with each loss formulation find 500 adversarial prompt embeddings from random initializations, and generate one image per prompt embedding to be classified in the same setting described in Section 4.1.

The results are presented in Table 4. We notice that, for concepts where the classifier has a high accuracy in classifying the images from the baseline model, as previously shown in Table 2, the $\widetilde{L}_2$ formulation performs marginally better than $\widetilde{L}_1$. For concepts whose classification accuracy is already low on the baseline model, the difference between the two loss formulations becomes negligible. This suggests when the baseline model can generate more 'accurate' images as perceived by the classifier, the outputs of its denoiser can also act as a more informative surrogate to aid the concept restoration process.

The RECORD algorithm discussed in this paper uses loss Equation [4] by default due to its marginal improvement in the attack performance. Consequently, this particular loss may result in the increased runtime of the RECORD algorithm, as well as requiring access to the baseline model $\epsilon_\theta$. These limitations, however, can be mitigated or avoided by switching to loss Equation [3] at the expense of a marginally poorer performance.

# C  ABLATION STUDY ON TOKENS LENGTHS

| Erased Concept | Prompt Length $S$ | Erasure Method | | | | | |
|---|---|---|---|---|---|---|---|
| | | ESD 2023a | FMN 2024a | AC 2023 | SPM 2024 | UCE 2023b | AdvUnlearn 2024b |
| van Gogh | 16 | 26.2 | 53.8 | 89.6 | 85.6 | 94.2 | 13.6 |
| | 32 | 40.0 | 69.6 | 92.4 | 91.8 | 95.4 | 29.4 |
| | 64 | **64.0** | **76.8** | **94.0** | **95.6** | **97.6** | **33.0** |

| Erased Concept | Prompt Length $S$ | Erasure Method | | | | | |
|---|---|---|---|---|---|---|---|
| | | ED 2024 | ESD 2023a | SalUn 2023 | SH 2024 | SPM 2024 | AdvUnlearn 2024b |
| Church | 16 | 34.8 | 58.0 | 52.8 | 4.6 | 89.4 | 42.4 |
| | 32 | **43.6** | **67.0** | **66.6** | **5.4** | **94.8** | **55.6** |
| | 64 | **61.2** | **75.2** | **71.4** | **8.6** | 92.2 | **57.0** |
| Garbage Truck | 16 | 18.8 | 26.2 | 43.0 | **1.6** | **69.0** | **63.8** |
| | 32 | **34.0** | **33.8** | **60.4** | 1.0 | **72.0** | 59.6 |
| | 64 | **40.8** | **38.8** | 58.0 | 1.0 | 66.4 | 50.0 |
| Parachute | 16 | 6.0 | 31.0 | 32.8 | **2.0** | 45.8 | 17.2 |
| | 32 | **10.0** | **36.6** | **35.4** | 1.0 | **55.8** | **26.2** |
| | 64 | **15.4** | **44.6** | **48.8** | 2.0 | **60.4** | **35.6** |
| Nudity | 16 | **3.6** | 62.2 | 5.2 | **23.6** | 54.8 | **48.4** |
| | 32 | 2.2 | **65.0** | **6.6** | **22.0** | 65.8 | **51.6** |
| | 64 | **2.4** | **70.6** | **9.0** | 21.2 | **69.0** | 39.2 |

Table 5: Attack success rate (%) of RECORD with different prompt length $S$.

| Number of Passes $N$ | Erasure Method | | |
|---|---|---|---|
| | ESD 2023a | AC 2023 | AdvUnlearn 2024b |
| 5 | 50.8 | 91.0 | 7.8 |
| 10 | 58.4 | 92.4 | **28.2** |
| 15 | **65.8** | **93.6** | 22.0 |
| 20 | **64.0** | **94.0** | **33.0** |

| Number of Passes $N$ | Prompt Length $S$ | Erasure Method | | |
|---|---|---|---|---|
| | | ESD 2023a | AC 2023 | AdvUnlearn 2024b |
| 80 | 16 | 33.6 | **91.4** | 30.4 |
| 40 | 32 | **45.8** | 91.2 | **33.8** |
| 20 | 64 | **64.0** | **94.0** | **33.0** |

Table 6: Attack success rate (%) of RECORD with different number of passes $N$ (left) and with fixed number of optimization steps (right) on models unlearned on van Gogh style.

This section presents an ablation study to examine the effect of two key parameters of RECORD: the number of tokens available for perturbation $S$ and number of passes on the token sequence $N$. The product of these two parameters, $N \times S$, corresponds to the total number of optimization steps, which linearly scales the algorithm's runtime. The rest of the parameter configuration is consistent with that of Section 4.2.

Tables 5 and 6 demonstrate a strong impact of both $N$ and $S$ on RECORD's performance. As expected, increasing these parameters generally leads to better performance. However, a notable finding is that even with $S = 16$, RECORD still achieves decent ASRs that often surpass those of P4D and UD with $S = 64$ in Table 3. For $S = 16$, RECORD also has a much shorter runtime as shown in Figure 3.

We also observed that reducing $S$ has a more pronounced negative effect on ASR than reducing $N$. This is evident in Table 6 (right), where a configuration with a smaller $S$ but a compensating increase in $N$, for maintaining a consistent total number of optimization steps, still resulted in a

decay in ASR. This decay, however, was less severe than a simple reduction in $S$ in Table 5 without a compensatory increase in $N$. This suggests the breadth of the search space of the adversarial tokens is more critical than the depth of the search, and choosing a larger $S$ is the preferred strategy for maximizing performance for the same compute cost.

# D ABLATION STUDY ON GRADIENT AND CANDIDATE TOKENS

| Gradient Token $J$ | Erasure Method | | | Candidate Token $K$ | Erasure Method | | |
|---|---|---|---|---|---|---|---|
| | ESD 2023a | AC 2023 | AdvUnlearn 2024b | | ESD 2023a | AC 2023 | AdvUnlearn 2024b |
| 1 | 60.2 | 93.8 | 16.4 | 1 | 17.4 | 65.8 | 2.4 |
| 8 | 61.4 | **94.6** | 30.2 | 8 | 54.6 | 93.0 | 8.4 |
| 16 | **64.0** | 92.2 | 30.0 | 16 | 54.2 | 90.4 | 17.8 |
| 32 | **65.2** | 92.8 | **30.6** | 32 | **61.2** | **95.2** | **29.8** |
| 64 | 64.0 | **94.0** | **33.0** | 64 | **64.0** | 94.0 | **33.0** |

Table 7: Combined tables showing van Gogh Gradient and Candidate Tokens.

| Prompt Length $S$ | Restoration Method Runtime$/s$ | | | |
|---|---|---|---|---|
| | P4D | UD | RECORD (J=64) | RECORD (J=8) |
| 8 | 2059±49 | 305±7 | **235±16** | **97±2** |
| 16 | 2050±34 | **429±7** | 464±18 | **191±2** |
| 32 | 2070±22 | **638±39** | 929±16 | **382±5** |
| 64 | 2079±22 | **1003±50** | 1859±15 | **832±7** |

Table 8: The mean and the standard deviation of the restoration method runtime are computed over 10 runs.

In this section, we conduct ablation studies on the gradient token number $J$ and candidate token number $K$ to justify the design choices as well as the performance and efficiency of the RECORD algorithm. The rest of the experimental setting is consistent to that of Section 4.2.

Our experiments demonstrate that increasing the number of gradient tokens $J$, which are used for linearised gradient estimation, yields only marginal performance improvements in terms of ASRs. In contrast, the number of candidate tokens $K$ for exact evaluation has a more significant impact on boosting ASRs.

This finding suggests it is possible to substantially accelerate the algorithm by significantly reducing $J$ to accelerate the algorithm. Notably, with $J = 8$, the runtime of the algorithm can be reduced by 60%, while having only a marginal relative drop in ASRs of $5 - 10\%$, or $3 - 4\%$ in absolute terms. Smaller $J$ also helps in reducing memory consumption, as gradient estimation through backpropagation with $J$ tokens corresponds to a major but now mitigable memory bottleneck.

# E  FIXED SEED ATTACKS

| Erased Concept | Restoration Method | Erasure Method | | | | | |
|---|---|---|---|---|---|---|---|
| | | **ED** 2024 | **ESD** 2023a | **SalUn** 2023 | **SH** 2024 | **SPM** 2024 | **AdvUnlearn** 2024b |
| Nudity | P4D 2023 | **3.4** | 84.8 | **21.2** | 4.2 | **100.0** | **27.1** |
| | UD 2023 | **1.7** | **87.3** | 19.5 | **11.0** | **100.0** | 22.0 |
| | **RECORD** | **1.7** | **98.3** | **30.5** | 9.3 | **100.0** | **41.5** |

Table 9: Attack success rate (%) of concept restoration methods in the fixed-seed setting.

To assess the performance of RECORD in the fixed-seed setting, we follow the experimental setup of UD (Zhang et al., 2023), where the target prompt-seed pairs are taken from the I2P dataset (Schramowski et al., 2023), which fixes the generation seed. In these experiments, we also similarly follow the previous works of using a NudeNet classifier (Bedapudi, 2019) and evaluates the optimized prompts on-the-fly: if a generated image is deemed unsafe, the optimization stops immediately.

Under this setting, we note RECORD achieves highly competitive performance in comparison with P4D and UD, especially on concept-erased models that are easier to attack, such as ESD, SalUn, and AdvUnlearn.

# F  ADVERSARIAL PROMPT TRANSFERABILITY

This section investigates the transferability of adversarial prompts generated by different concept erasure and restoration methods. Specifically, we examine whether prompts optimized on an unlearned SD1.4 model can successfully generate erased concepts on other models, such as SDXL and FLUX unlearned using ESD. This study assesses the generalizability and robustness of these prompts across different model architectures and explores the feasibility for an external adversary to use prompts optimized on a white-box open-source model to attack a different black-box model.

Our analysis uses a collection of adversarial prompts that can successfully generate erased concepts on their correspondingly unlearned SD1.4 models. Any prompts that fail to generate the erased concepts are not included in this study. As shown in Table 10, the adversarial prompts identified by the RECORD method exhibit greater transferability than those from P4D and UD, often by a significant margin. This suggests that the optimization strategy of RECORD produces prompts that are more robust and less model-specific.

We also observed that prompt transferability is highly dependent on both the specific erased concept and the target model. Adversarial prompts from SD1.4 generally transferred better to SDXL than to FLUX. This is likely due to the significant differences in training data and model architecture between SD1.4 and FLUX. Despite these differences, it is interesting to note that some level of transferability can still be preserved. This suggests a fundamental, underlying transferability of adversarial prompts in general, indicating that they retain some semantic meaning even when they appear as human-unreadable, gibberish-like strings.

| Erased Concept | Model | ESD | | | AC | | | AdvUnlearn | | |
|---|---|---|---|---|---|---|---|---|---|---|
| | | P4D | UD | RECORD | P4D | UD | RECORD | P4D | UD | RECORD |
| van Gogh | SDXL | 2.0 | **4.0** | **5.0** | **4.4** | 3.8 | **6.0** | 1.6 | **3.4** | **10.8** |
| | FLUX | **0.2** | 0.0 | **1.4** | 0.0 | **0.4** | **1.8** | **0.8** | 0.2 | **2.4** |

| Erased Concept | Model | ESD | | | SH | | | AdvUnlearn | | |
|---|---|---|---|---|---|---|---|---|---|---|
| | | P4D | UD | RECORD | P4D | UD | RECORD | P4D | UD | RECORD |
| Church | SDXL | **64.4** | 58.6 | **79.8** | 32.0 | **52.6** | **59.8** | 49.2 | **64.6** | **61.6** |
| | FLUX | **5.2** | 2.6 | **14.8** | **6.2** | 0.2 | **8.4** | **1.0** | **1.0** | **5.4** |
| Garbage Truck | SDXL | **85.0** | 75.2 | **89.4** | **25.0** | **32.6** | 21.2 | **55.6** | 51.4 | **64.8** |
| | FLUX | 0.0 | 0.0 | **0.8** | 0.0 | 0.0 | **0.2** | 0.0 | **0.4** | 0.0 |
| Parachute | SDXL | **56.2** | 55.2 | **63.4** | 2.6 | **21.6** | **23.0** | 23.8 | **37.4** | **46.4** |
| | FLUX | 0.0 | 0.0 | 0.0 | **0.2** | 0.0 | 0.0 | 0.0 | 0.0 | 0.0 |
| Nudity | SDXL | 61.8 | **68.0** | **77.6** | 11.8 | **75.4** | **22.0** | 49.8 | **74.2** | **66.0** |
| | FLUX | 18.0 | **18.4** | **38.0** | 5.4 | **16.4** | **9.6** | 8.2 | **11.6** | **22.4** |

Table 10: Attack success rate (%) of using successful adversarial prompts on SD1.4 on ESD-unlearned SDXL and FLUX.

# G SCALABILITY ON LARGE MODELS

| Candidate Selection Strategy | Erased Concepts | | | Average |
|:---:|:---:|:---:|:---:|:---:|
| | van Gogh | Church | Nudity | |
| Random switching | 17.4 | 86.0 | **23.6** | 32.5 |
| Interleaving | **26.6** | **92.4** | 17.6 | **34.8** |
| Blend $\alpha = 0.0$ | **19.0** | **92.4** | **26.6** | **35.4** |
| Blend $\alpha = 0.5$ | 14.2 | **91.0** | 17.6 | 31.4 |
| Blend $\alpha = 1.0$ | 7.2 | 89.2 | 15.0 | 28.4 |

Table 11: Attack success rate (%) of different candidate selection strategy for SDXL.

| Candidate Selection Strategy | Erased Concepts | | | Average |
|:---:|:---:|:---:|:---:|:---:|
| | van Gogh | Church | Nudity | |
| Random switching | 1.6 | **19.8** | **8.2** | **9.8** |
| Blend $\alpha = 0.0$ | **5.0** | **54.6** | **14.4** | **24.7** |
| Blend $\alpha = 1.0$ | **2.6** | 0.2 | 1.4 | 1.4 |

Table 12: Attack success rate (%) of different candidate selection strategy for FLUX.

We extend RECORD to work on larger models, namely SDXL (Podell et al., 2024) and FLUX (Labs et al., 2025). The major challenge is that SDXL and FLUX both use two separate text encoders for encoding input texts. This differs from SD1.4, where only one text encoder is used in its pipeline. This dual-encoder setup leads to difficulties in the candidate selection stage. For SDXL, both CLIP text encoders share a similar tokenizer with consistent token_id mapping, but with different embedding tables $E_1$, $E_2$ and different CLIP encoders. In this case, we propose three different strategies for handling candidate selection:

- **Random Switching**: At each optimization step, randomly choose one of the two text encoders for computing score and select the top-$K$ tokens.

$$\mathcal{V} = \text{Top}_K(E_i g_i), \quad \text{where } i \sim \text{Uniform}\{1, 2\},$$

where $g_i$ is the gradient of the corresponding embedding table $E_i$.

- **Interleaving**: Compute two separate scores and select the top-$K/2$ tokens with respect to each text encoder. In our experience, the number of overlapping candidate tokens are negligible compared to the size of $K$.

$$\mathcal{V} = \text{Top}_{K/2}(E_1 g_1) \cup \text{Top}_{K/2}(E_2 g_2).$$

- **Blend**: Compute a mixture of the scores from both text encoders and select the top-$K$ tokens.

$$\mathcal{V} = \text{Top}_K\Big((1 - \alpha)E_1 g_1 + \alpha E_2 g_2\Big),$$

where $\alpha \in [0, 1]$ is a tunable hyperparameter. When $\alpha = 0$ or $1$, this is equivalent to optimizing over only one text encoder.

We denote the embedding tables of CLIP-ViT/L (CLIPTextModel) and CLIP-ViT/G (CLIPTextModelWithProjection) as $E_1$ and $E_2$, respectively. The ASRs of the three strategies are presented in Table 11. We note that, for SDXL, using interleaving strategy or optimizing only on CLIP-ViT/L works the best. Example images can be found in Appendix L.

FLUX uses CLIP and T5 (Raffel et al., 2019) as its text encoders with two completely different tokenizers. Tokens of the two text encoders thus correspond to different strings of text. This is exacerbated by the non-bijective nature of the token-string mappings, i.e. original token $\rightarrow$ string $\rightarrow$ token $\neq$ original token. These make Interleaving and Blend not applicable, unless $\alpha \in \{0, 1\}$. Here we consider the embedding tables of CLIP and T5 encoder as $E_1$ and $E_2$. RECORD performs best on FLUX when only optimizing on the CLIP encoder (Table 12), which is mostly consistent with the behavior in SDXL.

## H EXPERIMENTS ON ADDITIONAL CONCEPTS AND CONCEPT ERASURE METHODS

In this section, we further expand the evaluation scope of our study to cover six additional concepts (tench, car, french horn, violence, illegal activity, Barack Obama) on two more concept erasure methods (SAFREE (Yoon et al., 2024), STEREO (Srivatsan et al., 2025)). The results continue to demonstrate the significantly superior performance of RECORD over other attack baselines.

| Erased Concept | Restoration Method | Erasure Method | | | |
|---|---|---|---|---|---|
| | | ESD 2023a | AdvUnlearn 2024b | SAFREE 2024 | STEREO 2025 |
| Car | P4D 2023 | **26.6** | **2.8** | **6.8** | 2.2 |
| | UD 2023 | 8.6 | 0.1 | 2.6 | **2.6** |
| | **RECORD** | 53.8 | 60.2 | 63.0 | 4.0 |
| French Horn | P4D 2023 | **17.4** | **5.8** | **0.8** | **0.4** |
| | UD 2023 | 2.3 | 0.0 | 0.0 | **0.4** |
| | **RECORD** | 70.6 | 87.8 | 94.0 | 1.0 |
| Violence | P4D 2023 | 28.6 | 7.6 | **25.8** | 7.0 |
| | UD 2023 | **37.8** | **10.9** | 11.8 | **13.2** |
| | **RECORD** | 67.2 | 57.2 | 73.0 | 14.6 |
| Drugs | P4D 2023 | **45.6** | **12.8** | 8.2 | **5.4** |
| | UD 2023 | 7.6 | 3.3 | **8.6** | 4.0 |
| | **RECORD** | 87.6 | 89.0 | 88.0 | 21.2 |
| Obama | P4D 2023 | **6.6** | 1.8 | 0.6 | **0.4** |
| | UD 2023 | 2.6 | **2.3** | **1.6** | 0.0 |
| | **RECORD** | 30.8 | 15.6 | 98.6 | 3.2 |
| Tench | P4D 2023 | **39.8** | 0.6 | 0.6 | 0.0 |
| | UD 2023 | 4.5 | **27.3** | **1.0** | **0.4** |
| | **RECORD** | 79.0 | 71.8 | 85.6 | 1.2 |

Table 13: Attack success rate (%) of additional concepts and concept erasure methods on SD1.4.

# I  LLM USAGE DECLARATION

Large language models have been occasionally used in this project for polishing writing, suggesting and applying bug fixes with significant human oversight, and for interfacing with computing infrastructure such as Slurm and Kubernetes.

# J  EMBEDDING-LEVEL ATTACK EXAMPLE IMAGES

In this section we present SD1.4 example images generated by different restoration methods, as well as from the embedding-level attack discussion in Appendix B

| Erased Concept | Restoration Method | Erasure Method | | | | | |
| | | ESD 2023a | FMN 2024a | AC 2023 | SPM 2024 | UCE 2023b | AdvUnlearn 2024b |
|---|---|---|---|---|---|---|---|
| van Gogh | No attack | | | | | | |
| | Embed attack 1 | | | | | | |
| | Embed attack 2 | | | | | | |

Table 14: Generated images of erased concepts using embedding-level attacks. Each image column of the same concept is generated using the same latent initialization. Models have unlearned the style of van Gogh paintings. Target prompt "a painting in the style of van Gogh"

| Erased Concept | Restoration Method | Erasure Method | | | | | |
| | | ESD 2023a | SH 2024 | SPM 2024 | SalUn 2023 | UCE 2023b | AdvUnlearn 2024b |
|---|---|---|---|---|---|---|---|
| Nudity | No attack | | | | | | |
| | Embed attack 1 | | | | | | |
| | Embed attack 2 | | | | | | |

Table 15: Generated images of erased concepts using embedding-level attacks. Each image column of the same concept is generated using the same latent initialization. Models have unlearned the concept of nudity. Target prompt asked for nudity.

## K  TOKEN-LEVEL ATTACK EXAMPLE IMAGES

| Erased Concept | Restoration Method | Erasure Method | | | | | |
|---|---|---|---|---|---|---|---|
| | | ESD 2023a | ED 2024 | SH 2024 | SPM 2024 | SalUn 2023 | AdvUnlearn 2024b |
| Church | No attack |  | | | | | |
| | Embed attack 1 | | | | | | |
| | Embed attack 2 | | | | | | |
| Garbage Truck | No attack | | | | | | |
| | Embed attack 1 | | | | | | |
| | Embed attack 2 | | | | | | |
| Parachute | No attack | | | | | | |
| | Embed attack 1 | | | | | | |
| | Embed attack 2 | | | | | | |

Table 16: The generated images of erased concepts using embedding-level attacks. Each image column of the same concept is generated using the same latent initialization. Target prompt "a photorealistic image of {object}"

| Erased Concept | Restoration Method | Erasure Method | | | | | |
|---|---|---|---|---|---|---|---|
| | | ESD 2023a | ED 2024 | SH 2024 | SPM 2024 | SalUn 2023 | AdvUnlearn 2024b |
| Church | P4D 2023 | | | | | | |
| | UD 2023 | | | | | | |
| | **RECORD** | | | | | | |
| Garbage Truck | P4D 2023 | | | | | | |
| | UD 2023 | | | | | | |
| | **RECORD** | | | | | | |
| Parachute | P4D 2023 | | | | | | |
| | UD 2023 | | | | | | |
| | **RECORD** | | | | | | |

Table 17: The generated images of erased concepts using token-level attacks. Each image column of the same concept is generated using the same latent initialization. Target prompt "a photorealistic image of {object}"

Table 18: The generated images of erased concepts using token-level attacks. Each image column of the same concept is generated using the same latent initialization. Target prompt asked for nudity.

| Erased Concept | Restoration Method | Erasure Method | | | | | |
|---|---|---|---|---|---|---|---|
| | | ESD 2023a | ED 2024 | SH 2024 | SPM 2024 | SalUn 2023 | AdvUnlearn 2024b |
| Nudity | P4D 2023 |  |  |  |  |  |  |
| | UD 2023 |  |  |  |  |  |  |
| | **RECORD** |  |  |  |  |  |  |

## L  EXAMPLE IMAGES FROM LARGER MODELS

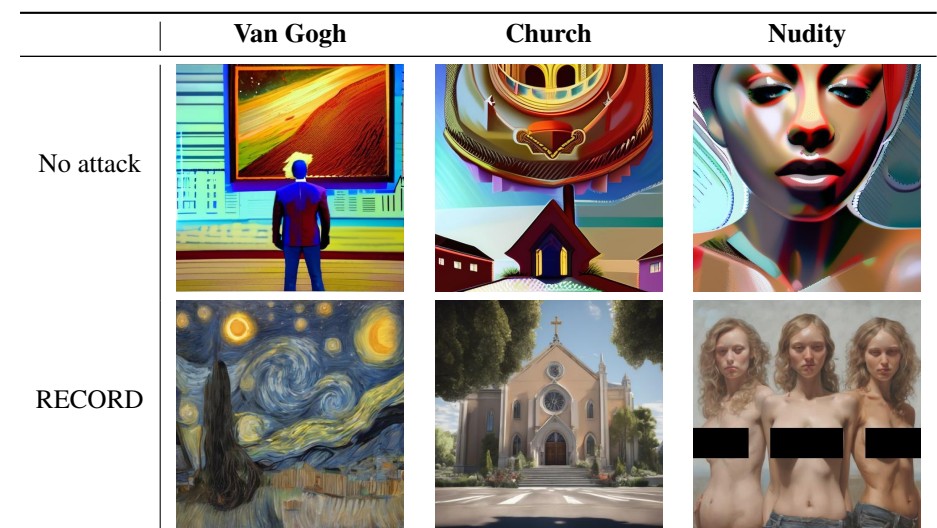

| | Van Gogh | Church | Nudity |
|---|---|---|---|
| No attack | | | |
| RECORD | | | |

Table 19: Images generated by ESD erased FLUX on van Gogh paintings, churches and nudity.

| | Van Gogh | Church | Nudity |
|---|---|---|---|
| No attack | | | |
| RECORD | | | |

Table 20: Images generated by ESD erased SDXL on van Gogh paintings, churches and nudity.

