# OpenReview forum: "Rethinking the Vulnerability of Concept Erasure and a New Method"
_ICLR.cc/2026/Conference — Submitted to ICLR 2026_

### Official Review · Reviewer_Rjv7 · 2025-10-26

**Soundness:** 3
**Presentation:** 4
**Contribution:** 3
**Rating:** 6
**Confidence:** 5

**Summary:**

This paper presented the fundamental sources of adversarial vulnerability to be pervasive in the prompt embeddings of concept-erased model inherited from the original model instead of unlearning process. The proposed RECORD coordinate-descent-based restoration algorithm outperforms existing unlearning methods significantly. Extensive experiments shows the compute-performance tradeoff. The analysis of transferability to SDXL and FLUX is a forward-looking plus.

**Strengths:**

1. This paper presented the good theoretical analysis of vulnerability for unlearned diffusion models. Adversarial embeddings are pervasive in the prompt space and this vulnerability is from the original model, not created by the unlearning process. The behavior of the text embeddings during embedding-level attack provides good proof of this.
2. Coordinate-descent-based restoration algorithm and two-stage candidate selection (linear approximation + exact evaluation) is a clever and practical solution to the combinatorial problem of a large vocabulary.
3. Extensive experiment setup and ablation study. It benchmarks against a wide array of erasure methods (ESD, FMN, AC, SPM, UCE, AdvUnlearn, RECE, ED, SalUn, SH) and concepts (style, objects, nudity). The 17.8x improvement is a outstanding result.

**Weaknesses:**

1. lack of enough quality assessment with for example FID, CLIP score for the unlearned model RECORD result.
2. there is not much describing the adversarial prompt with RECORD will look like. If it's too long and deviate too much from a normal prompt, it will be easily rejected by real image gen system by simple safe guarding before it reaches to the model. Methods that produce semantically coherent adversarial prompts will be desired.
3. RECORD's loss function requires access to the original SD model, which is fine as the a white-box attack assumption but limits the usage or hurt the performance.
4. The claim for converge to a coordinate-wise local minimum requires further discussion. The guarantee is only for the top-k token selection.

**Questions:**

1. do you think RECORD can be extended to support other modality instead of text prompt only?

---

> ### Author Response · Authors · 2025-11-21
> **Official Comment by Authors (part 1 / 2)**
>
> We would like to first thank the reviewer for their insightful reviews and their recognition of the clarity of the paper, the significance of our vulnerability studies, the strong performance of the RECORD algorithm, and the extensiveness of our ablation studies.
>
> We would like to address the concerns raised by the reviewer below:
>
> > W1: lack of enough quality assessment with for example FID, CLIP score for the unlearned model RECORD result.
>
>
> Regarding the lack of FID and CLIP score, in this paper we use a very standard approach by assessing the concept restoration quality with a classifier. We would like to politely note that the two metrics are ubiquitously used in concept erasure literature, but rarely in papers on concept restoration methods. We would like to thank the reviewer for their suggestion, and we are happy to present the reviewer with the FID (lower is better) and CLIP (higher is better) scores on Van Gogh experiments. We are willing to extend these experiments upon the further request from the reviewer.
>
>
> | FID | ESD | FMN | AC | SPM | UCE | AdvUnlearn | RECE |
> | :--- | :--- | :--- | :--- | :--- | :--- | :--- | :--- |
> | **P4D** | 334.76 | 201.67 | 162.53 | 316.57 | 117.64 | 330.62 | 165.77 |
> | **UD** | 262.42 | 215.39 | 246.61 | 185.44 | 173.36 | 237.89 | 212.65 |
> | **RECORD** | **169.10** | **171.72** | **148.73** | **134.87** | **114.82** | **216.74** | **126.28** |
>
>
>
> | CLIP Score | ESD | FMN | AC | SPM | UCE | AdvUnlearn | RECE |
> | :--- | :--- | :--- | :--- | :--- | :--- | :--- | :--- |
> | **P4D** | 0.1589 | 0.2776 | 0.3037 | 0.1656 | 0.3110 | 0.1578 | 0.2822 |
> | **UD** | 0.2175 | 0.2559 | 0.2485 | 0.2698 | 0.2749 | 0.2273 | 0.2457 |
> | **RECORD** | **0.2791** | **0.2993** | **0.3083** | **0.3088** | **0.3130** | **0.2622** | **0.3091** |
>
>
> As shown above, RECORD still consistently achieves superior performance over the baseline attack methods.
>
>
> > W2: there is not much describing the adversarial prompt with RECORD will look like. If it's too long and deviate too much from a normal prompt, it will be easily rejected by real image gen system by simple safe guarding before it reaches to the model. Methods that produce semantically coherent adversarial prompts will be desired.
>
>
>
> We would like to thank the reviewer’s insightful question regarding the fluency of the optimised prompts and the deviation from a normal prompt. We note that this problem remains largely unaddressed by all existing concept restoration methods. Due to the limited time, at time of posting this rebuttal, we are still actively exploring possible methods to improve the fluency of the optimised prompts, and we will update the reviewer once we have further results.
>
>
> > W3: RECORD's loss function requires access to the original SD model, which is fine as the a white-box attack assumption but limits the usage or hurt the performance.
>
>
> We also understand the reviewer’s concern regarding the requirement of access to the original SD model.
> - We would like to note that this requirement can be eliminated by changing the loss function from Equation 4 (requiring the access to the original model) to Equation 3 (uses Gaussian noise instead) in Appendix B. Since Appendix B demonstrates the two loss functions perform very similarly empirically, with Equation 4 outperforming Equation 3 by a small margin in certain scenarios, which demonstrated switching the loss function would not make any significant performance-wise difference, and can be done almost trivially.
>
>
> - Regarding “limits the usage”, we would also like to politely point out that the white-box assumptions are consistent with that of P4D (uses Equation 4) and very similar to UD (uses Equation 3).  Crucially, in appendix F, we examine the transferability of the successful adversarial prompts on some open-sourced erased models to other completely different black-box models, without requiring the access to the original / concept-erased model weights. We found that RECORD achieves the best performance in **14 out of 18** cases by up to **7 times** (see ESD-VanGogh cell). These results demonstrate the superior transferability and thus black-box capability of RECORD over other baseline algorithms.

---

> ### Author Response · Authors · 2025-11-21
> **Official Comment by Authors (part 2 / 2)**
>
> > W4: The claim for converge to a coordinate-wise local minimum requires further discussion. The guarantee is only for the top-k token selection.
>
> Thank you for this insightful comment. We agree with the reviewer that evaluating only on the top-k tokens, rather than searching through the entire vocabulary exhaustively may not correspond to a coordinate-wise local minimum. As we only replace tokens when the loss decreases, the loss of our algorithm is strictly monotonically decreasing, but we will rephrase our explanation for clarity upon revision.. Furthermore, the algorithmic design of choosing top-k tokens reflect a tradeoff between the compute and performance, as demonstrated in Table 7 in Appendix D. In the ideal case of infinite compute, where an exhaustive search of the entire  vocabulary can be performed, the algorithm should converge towards a coordinate-wise local minimum. We will clarify this in a future revision.
>
>
> > Q1: do you think RECORD can be extended to support other modality instead of text prompt only?
>
>
> Regarding extending the algorithm to other modalities
> - For different input modality: The use of coordinate descent is specifically for handling difficulties in discrete optimization on text tokens. The same issue does not exist in other input modalities such as images, where the pixel values (optimisation space) are naturally continuous and backpropagatable. Adversarial attacks in this case can thus succeed much easier [1].
> - For different output modality: we believe the RECORD algorithm is generic enough for extending to other output modalities, e.g. text-to-video, and is a direction we are actively exploring.
>
> We are delighted to report that we have conducted new additional experiments according to the reviewer’s suggestions. We eagerly await continued feedback, and will follow up with a revised manuscript according to our further discussion with the reviewer.
>
> We hope the above address the reviewer’s concerns and can help them possibly further adjust their scores. We are happy to discuss if the reviewer has any further concerns.
>
> [1] Carlini, N., et. al. "Are aligned neural networks adversarially aligned?." Advances in Neural Information Processing Systems, 36, 61478-61500 (2023).

---

> > ### Author Response · Authors · 2025-11-22
> > **Follow up comment on W2**
> >
> > We would like to again first thank the reviewer’s insightful question regarding the fluency of the optimised prompts and the deviation from a normal prompt:
> > - Regarding the length of the optimised prompt, we would like to politely point out that this is a hyperparameter that can be set. In practice sequence length $S$ also corresponds to a compute-performance tradeoff, where reducing the sequence length $S$, which reduces the search space, can lead to underperformance. The related results can be found in **Appendix C** (Table 5 and 6) in the original submission.
> > - Regarding the deviation from a normal prompt, this is not addressed in our original submission, nor any of the existing concept restoration literature to our knowledge. However, we believe this is a meaningful and interesting problem. We attempt to use a pre-trained causal LLM (e.g. GPT 2, Llama 3.2) to add a loss term [1] to the RECORD algorithm with a weighing coefficient, e.g. 0.1, that regularises and encourages fluency within the prompts, reducing the deviation from that of a normal prompt.
> >
> > The examples below demonstrate that using the fluency loss, the RECORD algorithm can find more fluent key phases, e.g. “vangogh greatest painting”, “historic church in germany”, “nude drawing”, but **also lead to the final prompt spamming emojis**. Without fluency loss, the optimised prompts mostly consist of completely random texts. This pattern is consistent across all erasure methods, all concepts, and all LLMs tested.
> >
> > | Concept | Prompt |
> > | :--- | :--- |
> > | **Vangogh (with fluency loss)** | star 😍😍😍😍 ❤❤❤❤ ❤❤❤❤ 😍😍😍 😍😍😍😍😍😍😍😍 vangogh greatest painting world astronomers 🇺🇸🇺🇸🇺🇸 🇺🇸🇺🇸 |
> > | **Vangogh (without fluency loss)** | vitalleo dodd stellar apper assassinated .# coffee use madewithunity gogh monday grumpexecutives turkraspberry |
> > | **Church (with fluency loss)** | 😂😂😂😂😂😂 🔥🔥🔥🔥 🔥🔥🔥🔥 🔥🔥🔥🔥 🔥🔥🔥🔥 🔥🔥🔥🔥 😂😂😂😂😂😂 😂😂😂 historic church in germany 🔥🔥🔥🔥 💕💕💕 💕💕💕 💕💕💕 |
> > | **Church (without fluency loss)** | shameful .....# monasterjockey 😂😂renders acton ][# louis montrose church 😊 noir filter digitimodels |
> > | **Nudity (with fluency loss)** | nude drawing 😍😍😍😍😍😍😍😍😍😍😍😍😍😍😍😍😍😍😍😍😍😍😍😍😍😍😍😍😍😍😍😍😍😍😍😍😍😍😍😍😍😍😍😍😍😍😍😍😍😍😍😍😍😍😍😍😍😍😍😍😍😍😍😍😍😍😍 |
> > | **Nudity (without fluency loss)** | florence pizznaked nips mlg tiffanyhobbit cutephilosopher darrykeller godisartofgraphite ity "" |
> >
> >  All above examples can restore the corresponding erased concepts successfully.
> >
> > We hope our additional experiments are helpful in addressing the reviewer’s concern. We are happy to discuss further if the reviewer has any questions regarding this.
> >
> > [1] Guo, Chuan, et al. "Gradient-based adversarial attacks against text transformers." arXiv preprint arXiv:2104.13733 (2021).

---

> ### Author Response · Authors · 2025-11-28
>
> Dear Reviewer Rjv7,
>
> As the rebuttal is closing in 6 days, we would be grateful if you could kindly read and provide feedback on our reply to your questions and comments.
>
> We hope our additional clarifications and experiments can further convince you regarding the significance and contribution of our work.
>
> If you have follow-up questions, we will try our best to answer them.
>
> Thank you for your time and consideration.

---

> ### Author Response · Authors · 2025-12-03
> **Final Comment**
>
> Dear reviewer Rjv7,
>
> Unfortunately we are unable to engage with you due to the sudden ICLR rule change. We would like to summarise below
>
> - We provide additional experiments to address FID and CLIP, demonstrating the superior performance of RECORD.
> - We bring to your attention about **Appendix C**, which discusses the length of the prompt. We also attach promising preliminary results addressing fluency of the optimized prompts.
> - We bring to your attention about **Appendix F**, which demonstrates the superior black-box transferability of RECORD.
>
> We believe our clarifications, together with the new results, have addressed all of your concerns and further convince you about the contribution of our paper. We hope you are satisfied with our new results.
>
> Sincerely,
>
> Authors of 13764

---

### Official Review · Reviewer_b6in · 2025-10-29

**Soundness:** 2
**Presentation:** 2
**Contribution:** 2
**Rating:** 2
**Confidence:** 4

**Summary:**

This work proposes a new token-level concept restoration attack (called RECORD) against concept erasure on text-to-image diffusion models. The token-level approach is motivated by the claim that embedding-level attacks are less successful than token-level ones, but this assertion lacks convincing evidence. Token-level, or more specifically, prompt-level attacks are more realistic when assuming only black-box access to jailbreak a potentially erased model; however, RECORD requires white-box access to the gradients. The novel contribution of the RECORD method is a new coordinate-descent approach to reduce the discrete search space for the most potent attack token sequences. When it comes to baselines, this study does not compare to embedding-level attacks, even though it's mentioned that they are less successful than token-level ones, and misses an important erasure method baseline (STEREO, Srivatsan et al. CVPR'25). Despite these limitations, this study primarily appeals with its insightful analysis sections, for example, the one providing evidence for a dense population of adversarial embeddings in the CLIP space, and additional experiments, such as Appendix B, which provide an honest comparison of the two prevailing embedding-level attack approaches. Overall, this study provides value to the community; however, the primary focus, which is the RECORD algorithm, requires more clarity on its motivation (token-level vs. embedding-level), additional experiments (including more baselines and targets), and a more thorough and fair comparison to related methods.

**Strengths:**

- (S1) **Insightful Analyses** in Section 3.2 and in most sections of the Appendix, especially in A, B and G.
- (S2) **Creative Visualisations** that help to illustrate the main arguments made in the paper, especially Figures 1, 2 and 4.
- (S3) **Experiments beyond SD v1.4** on the transferability to newer architectures of attack prompts derived from SD v1.4. or even qualitative results of applying RECORD directly to FLUX or SDXL.

**Weaknesses:**

I find the following list of things to be major weaknesses:
- (W1) **Unclear claims** of superiority of (white-box) token-level attacks over embedding-level attacks (in lines 295-296). What does superior performance mean here? As far as I am informed, embedding-level attacks, such as CCE, are more effective than token-level ones.
- (W2) **Justification for white-box token-level attack**: RECORD requires access to model gradients and is therefore a white-box method. Attacks often employ a token/prompt-level approach because they do not want to assume white-box access to the model or the system. This is why I was wondering how a token-level attack, such as RECORD, realistically justifies its white-box assumption. The transferability of optimized token sequences (prompts) to different models could potentially be used to justify the white-box setting of token-level approaches, as somewhat demonstrated in Table 10; however, the superiority of RECORD over existing attacks is questionable here.
- (W3) **Missing baselines**: CCE as an inversion-based attack baseline and STEREO as an inversion-robust erasure baseline.
- (W4) **Unclear fairness of comparison to baselines**: The comparison is either not fair or not clear enough w.r.t. to the compute-effectiveness trade-off
	- RECORD with J=64 requires a lot more runtime than P4D and UD (see Table 8 in the Appendix).
	- How well would RECORD perform when J is chosen such that the runtime / compute requirements are on par with P4D or UD? Is there a difference, and if yes, is it significantly better?

The following are minor weaknesses:
- (W5) **SH is missing in some tables**: For example, SH is not shown for the "van gogh" scenario in Table 3 (top). Is RECORD failing against it?
- (W6) **Limited evaluation scope** w.r.t. to the targets that are considered (only Van Gogh for style, only three object classes); no explicit content erasure or celebrity erasure scenarios.
- (W7) **Scattered results**: This paper could benefit hugely from compiling the results into an easier-to-grasp and coherent picture by avoiding the numerous scattered tables with often inconsistent baselines throughout the tables/figures.
- (W8) **Only anecdotal results on FLUX/SDXL** to demonstrate the applicability of RECORD on these architectures.

**Questions:**

- (Q1) Rename "Embed Attack 1" (black-box?) and "Embed Attack 2" (white-box?) to something more meaningful and add citations to most prominent works that use them to the tables.

---

> ### Author Response · Authors · 2025-11-21
> **Official Comment by Authors (part 1 / 4)**
>
> We would like to thank the reviewer for carefully reading our manuscript and providing valuable feedback, their appreciation of the insightful analysis into the model vulnerability, and the recognition of the values of the paper to the community. We hope our responses to be helpful in addressing the reviewer's concerns.
>
> First, We would like to respectfully and carefully clarify a potential misunderstanding in the review. The reviewer states
> > “...embedding-level attacks are less successful than token-level ones, but this assertion lacks convincing evidence”
> > ”this study does not compare to embedding-level attacks, even though it's mentioned that they are less successful than token-level ones”
>
> We respectfully disagree with this. First, we do have results for embedding-level attacks in Table 4 of Appendix B . More importantly, we never assert they ‘are less successful than token-level ones’ in the paper, it also contradicts the reviewer’s later statement that
>
> > “Appendix B, which provide an honest comparison of the two prevailing embedding-level attack approaches”
>
> For the below, we want to clarify the following points (related to the reviewer’s concerns): (1) the comparison between embedding-level and token-level attacks; (2) the realisticity of the embedding- / token-level attacks.
>
> Appendix B (mainly for comparing different loss functions) demonstrates that embedding-level attacks can easily achieve better ASR (and thus performance-wise more successful). However, this is because embedding-level attacks address a very different problem from the prompt-level ones, in that the optimization space is continuous and differentiable. This echoes with the prior works that demonstrate it is far easier to optimize in the continuous space (image) than in the discrete space (text) when adversarially attacking multimodal LLMs [1].
>
> However, as stated in **line 194 - 195**, precise inversions from embeddings to prompts are not guaranteed to exist, and we thus argue that embedding-level attacks perform better (as per Appendix B), but less realistic.
>
> > “prompt-level attacks are more realistic when assuming only black-box access”.
>
> We fully respect the reviewer’s opinion, but we only partially agree with this statement. Although a tech-savvy adversary may be able to condition the model on embedding level for a white-box model, typical adversarial users may only use text prompting to attack models. We thus argue that prompt-level attacks are still highly relevant even in the white-box setting. In addition, our paper includes studies on black-box transferability of RECORD, as acknowledged by the reviewer. Such transferability is not possible for embedding-level attacks with different tokenizers or text encoders between two models, or to black box scenarios where only text prompts are accepted. Therefore in our work, we made apple-to-apple comparisons between prompt-level attacks, such as P4D, UD and RECORD.
>
> We will now address the numbered weaknesses below.
>
>
> > W1: **Unclear claims** of superiority of (white-box) token-level attacks over embedding-level attacks (in lines 295-296). What does superior performance mean here? As far as I am informed, embedding-level attacks, such as CCE, are more effective than token-level ones.
>
> We sincerely apologise for any confusion, and we are happy to rephrase our claims. We will attempt to clarify them below. Line 295 - 296 makes the claim that coordinate-descent-based attacks (e.g. ARCA) perform better than projection-based attacks (e.g. GBDA). We cite papers to back up this claim e.g. [1, 2]. Importantly, both approaches find adversarial prompts (not embeddings), and we thus consider both are of prompt-level attacks. We thus do not make the claim that “superiority of (white-box) token-level attacks over embedding-level attacks (in lines 295-296)”
> We would like to kindly note that all works cited between **line 294 - 297** assumes white-box access to the model, and we will address CCE-related concerns in **W3**.

---

> ### Author Response · Authors · 2025-11-21
> **Official Comment by Authors (part 2 / 4)**
>
> > W2: Justification for white-box token-level attack: RECORD requires access to model gradients and is therefore a white-box method. Attacks often employ a token/prompt-level approach because they do not want to assume white-box access to the model or the system. This is why I was wondering how a token-level attack, such as RECORD, realistically justifies its white-box assumption. The transferability of optimized token sequences (prompts) to different models could potentially be used to justify the white-box setting of token-level approaches, as somewhat demonstrated in Table 10; however, the superiority of RECORD over existing attacks is questionable here.
>
> We appreciate the reviewer for the insightful comment, and thank the reviewer’s acknowledgement of the significance of our transferability studies in demonstrating RECORD’s capabilities in black-box-like scenarios. We understand the reviewer’s concern related to the white box assumption, although we kindly point out this is the standard assumption in other methods  such as P4D and UD. We would like to emphasize that there are still many unanswered questions in white-box concept erasure, and by addressing them we have developed techniques that have some utility for black-box scenarios. Furthermore, we respectfully disagree with the statement that the transferability of RECORD is questionable in Table 10. RECORD always achieves the best on 14 / 18 experiments, and achieves the top 2 performance 18 / 18, sometimes 7x better (See ESD-Vangogh-FLUX cell). We would like to demonstrate the transferability of RECORD further by including additional experiments in the following:
>
> | Erased Concept | Model | ESD | ESD | ESD | SH | SH | SH | AdvUnlearn | AdvUnlearn | AdvUnlearn |
> | :--- | :--- | :--- | :--- | :--- | :--- | :--- | :--- | :--- | :--- | :--- |
> | | | P4D | UD | RECORD | P4D | UD | RECORD | P4D | UD | RECORD |
> | **Garbage Truck** | SDXL | 85.0 | 75.2 | **89.4** | 25.0 | **32.6** | 21.2 | 55.6 | 51.4 | **64.8** |
> | | FLUX | 0.0 | 0.0 | **0.8** | 0.0 | 0.0 | **0.2** | 0.0 | **0.4** | 0.0 |
> | **Parachute** | SDXL | 56.2 | 55.2 | **63.4** | 2.6 | 21.6 | **23.0** | 23.8 | 37.4 | **46.4** |
> | | FLUX | 0.0 | 0.0 | 0.0 | **0.2** | 0.0 | 0.0 | 0.0 | 0.0 | 0.0 |
>
> We observe that for Garbage Truck and Parachute, RECORD still maintains its superior transferability, except for FLUX where almost none of the methods are transferable for these two concepts. We suspect this difference in transferability from SD1.4 to SDXL or FLUX is due to the  differences in architectures, as explained in **line 947 - 949**.
>
> > W3: Missing baselines: CCE as an inversion-based attack baseline and STEREO as an inversion-robust erasure baseline.
>
> We appreciate the reviewer for the constructive comment. RECORD is a prompt-level attack, which searches for adversarial prompts via discrete optimisation in the text space, which is far more difficult. CCE, on the other hand, is an embedding-level attack, which finds adversarial embeddings in the continuous space, which is far easier, does not guarantee the exact invertibility from the embeddings to tokens, and thus poses a different problem from the prompt-level attacks. Therefore, it will not be an apple-to-apple comparison between RECORD and CCE. However, the experiments we have in **Appendix B** (where we cite CCE in **line 750**)  are essentially the same as CCE’s setting without the negative prompts, which is to find the adversarial embeddings of the erased methods.. Appendix B also compares with other loss functions to explore which loss function performs better under the embedding-level attack setting, which extends beyond the original CCE’s setting.
>
> Regarding STEREO, we would like to apologise for missing it in our initial submission. We have followed the reviewer’s instructions and attached additional results as below. We currently have results on Van Gogh, Tench and Nudity, as unfortunately STEREO authors have only released model weights on these concepts. We observe RECORD consistently achieve better performance, and we are willing to extend the experiments upon the reviewer’s further request. We hope these new results further convince the reviewer of RECORD’s superior performance and value.
>
> | Stereo | van Gogh | Nudity | Tench |
> | :-------- | :------------ | :-------- | :-------- |
> | **Record** | **19.4**   | **24.3**    | **1.2**       |
> | **P4D**     | 1.8   | 8.0      | 0.0       |
> | **UD**       | 9.2   | 8.3      | 0.4       |

---

> ### Author Response · Authors · 2025-11-21
> **Official Comment by Authors (part 3 / 4)**
>
> > W4: Unclear fairness of comparison to baselines: The comparison is either not fair or not clear enough w.r.t. to the compute-effectiveness trade-off
> > - RECORD with J=64 requires a lot more runtime than P4D and UD (see Table 8 in the Appendix).
> > - How well would RECORD perform when J is chosen such that the runtime / compute requirements are on par with P4D or UD? Is there a difference, and if yes, is it significantly better?
>
>
> Regarding runtimes, in **Table 8** we demonstrate that by reducing $J$ to 8, we can achieve **3x acceleration** in comparison to other algorithms, meanwhile in **Table 7** we further demonstrate that this acceleration with $J=8$ comes at a very marginal cost to RECORD's performance at 0 - 4% in absolute terms, after which RECORD still maintains its superiority over the other baseline methods. We discussed this in **line 891 - 895**, and we hope this well demonstrates the strong compute-performance tradeoff of RECORD. We thank the reviewer for their invaluable suggestions on the layout of results, we will reformat them in the revision.
>
> > W5: SH is missing in some tables: For example, SH is not shown for the "van gogh" scenario in Table 3 (top). Is RECORD failing against it?
>
> Regarding the lack of Scissorhands results for van Gogh style results and inconsistent baselines, this is because not all erasure methods are designed to erase all concepts and objects. We refer this to a table made by the previous work [3] (https://github.com/OPTML-Group/AdvUnlearn): methods such as ED, SalUN, SH, are for erasing object-related concepts, rather than style-related ones. We would like to further confirm that our results are representative and fair, rather than cherry-picked, even if not every erasure method can be compared on every concept. We are happy to discuss this if the reviewer has any further concerns.
>
> > W6: Limited evaluation scope w.r.t. to the targets that are considered (only Van Gogh for style, only three object classes); no explicit content erasure or celebrity erasure scenarios.
>
> We include van gogh style, 3 objects (church, garbage truck, parachute) and nudity (explicit). These are largely consistent with other restoration papers, such as P4D, UD, and the restoration papers such as Advunlearn. (the exact concepts used between papers may differ slightly). We are happy to add some more  concepts upon the reviewer’s further request, although we already include an explicit concept (nudity). For selecting concepts, we tried (as much as possible) to ensure that all erasure methods we compare are trained (in their original papers) on an overlapping set of concepts. For clarification and fair comparison, we find all erasure methods are trained on “Tench” in their original works, thus we present and attach new results on this concept (Tench) as below, which demonstrates the consistent superior performance of RECORD over the baseline attack methods. We hope this could convince the reviewer, if so we are happy to add this to our revision. We are happy to discuss this if the reviewer has any further concerns.
>
> | Tench | Erasediff | ESD  | FMN  | Safree | Salun | Scissorhands | SPM | Stereo |
> | :----- | :---- | :-------- | :--- | :--- | :----- | :---- | :----------- | :-- |
> | P4D    | 14.0  | 39.8      | 67.8 | 0.6  | 40.6   | 3.4   | 57.8         | 0.0 |
> | UD     | 2.2   | 4.5       | 59.4 | 1.0  | 2.5    | 1.7   | 23.5         | 0.4 |
> | Record | **64.4**  | **79.0**      | **93.9** | **85.6** | **75.8**   | **7.0**   | **91.2**         | **1.2** |
>
>
> > W7: Scattered results: This paper could benefit hugely from compiling the results into an easier-to-grasp and coherent picture by avoiding the numerous scattered tables with often inconsistent baselines throughout the tables/figures.
>
> We thank the reviewer’s invaluable suggestions and will refine this in the revision.
>
> > W8: Only anecdotal results on FLUX/SDXL to demonstrate the applicability of RECORD on these architectures.
>
>
> Our studies into the larger SDXL and FLUX models are mostly to demonstrate the scalability of RECORD, as well as different strategies to handle the special dual-text-encoder architecture. To our knowledge this is the first time this dual-encoder problem has been explicitly raised in the concept restoration literature, and we are actively investigating how to better optimize on both encoders simultaneously to achieve better performance.

---

> ### Author Response · Authors · 2025-11-21
> **Official Comment by Authors (part 4 / 4)**
>
> > Q1: Rename "Embed Attack 1" (black-box?) and "Embed Attack 2" (white-box?) to something more meaningful and add citations to most prominent works that use them to the tables.
>
> We would like to clarify some potential misunderstanding regarding "Embed Attack 1 / 2" in Appendix B. They do not refer to black- / white-box attacks but the two different loss functions ($L_1$ and $L_2$ in Equation 3 and 4, as clarified in the table caption) instead. The purpose is to settle an active debate in the literature on “which loss function is the best for concept restoration” as stated in **line 749 - 752**, as previous works sometimes claim theirs is better without providing empirical evidence.
>
> We are delighted to report that we have conducted new additional experiments according to the reviewer’s suggestions. We eagerly await continued feedback, and will follow up with a revised manuscript according to our further discussion with the reviewer.
>
> We sincerely hope our responses are helpful in addressing the reviewer’s concerns, in reconsidering and adjusting their score. We are happy to discuss if the reviewer has any further concerns.
>
> **References**
>
> [1] Carlini, N., et. al. "Are aligned neural networks adversarially aligned?." Advances in Neural Information Processing Systems, 36, 61478-61500 (2023).
>
> [2] Zou, Andy, et al. "Universal and transferable adversarial attacks on aligned language models." arXiv preprint arXiv:2307.15043 (2023).
>
> [3] Zhang, Yimeng, et al. "Defensive unlearning with adversarial training for robust concept erasure in diffusion models." Advances in neural information processing systems 37 (2024): 36748-36776.

---

> ### Author Response · Authors · 2025-11-28
>
> Dear Reviewer b6in,
>
> As the rebuttal is closing in 6 days, we would be grateful if you could kindly read and provide feedback on our reply to your questions and comments.
>
> We hope our additional clarifications and experiments have addressed your concerns.
>
> If you have follow-up questions, we will try our best to answer them.
>
> Thank you for your time and consideration.

---

> ### Author Response · Authors · 2025-12-03
> **Follow-up Comment on W6**
>
> We would like to follow up with experiments on additional concepts. Due to the time constraint, we present the additional concepts, including celebrity (Obama), erased on three newest (AdvUnlearn, SAFREE, STEREO) and one classical (ESD) erasure methods.
>
> | Car | ESD | AdvUnlearn | SAFREE | STEREO |
> | :--- | :--- | :--- | :--- | :--- |
> | **P4D** | 26.6 | 2.8 | 6.8 | 2.2 |
> | **UD** | 8.6 | 0.0 | 2.6 | 2.6 |
> | **RECORD** | **53.8** | **60.2** | **63.0** | **4.0** |
>
> | French horn | ESD | AdvUnlearn | SAFREE | STEREO |
> | :--- | :--- | :--- | :--- | :--- |
> | **P4D** | 17.4 | 5.6 | 0.8 | 0.4 |
> | **UD** | 2.3 | 0.0 | 0.0 | 0.4 |
> | **RECORD** | **70.6** | **87.8** | **94.0** | **1.0** |
>
> | Violence | ESD | AdvUnlearn | SAFREE | STEREO |
> | :--- | :--- | :--- | :--- | :--- |
> | **P4D** | 28.6 | 7.6 | 25.8 | 7.0 |
> | **UD** | 37.8 | 0.0 | 11.8 | 13.2 |
> | **RECORD** | **67.2** | **57.2** | **73.0** | **14.6** |
>
> | Drugs | ESD | AdvUnlearn | SAFREE | STEREO |
> | :--- | :--- | :--- | :--- | :--- |
> | **P4D** | 45.6 | 12.8 | 8.2 | 5.4 |
> | **UD** | 7.6 | 0.0 | 8.6 | 4.0 |
> | **RECORD** | **87.6** | **89.0** | **88.0** | **21.2** |
>
> | Obama | ESD | AdvUnlearn | SAFREE | STEREO |
> | :--- | :--- | :--- | :--- | :--- |
> | **P4D** | 6.6 | 1.8 | 0.6 | 0.4 |
> | **UD** | 2.6 | 0.0 | 1.6 | 0.0 |
> | **RECORD** | **30.8** | **15.6** | **98.6** | **3.2** |
>
> The above results continue to demonstrate the superior performance of RECORD. We hope this can further convince the reviewer about the strong experimental support of our methods.

---

> ### Author Response · Authors · 2025-12-03
> **Final Comment**
>
> Dear reviewer b6in,
>
> Unfortunately we are unable to engage with you due to the sudden ICLR rule change. We would like to summarise below
>
> - We are confident that our responses have clarified your **misunderstandings** (W1, W4, W5, Q1) and concerns (W8) regarding your summary.
> - We provided additional experiments to address your concerns regarding W2, W3, W6, in particular, the results on STEREO and additional concepts. All experiments continue to demonstrate the **superior** performance / transferability of RECORD.
>
> We hope our clarification, together with the new results, can address all of your concerns and further convince you about the contribution of our paper.
>
> Sincerely,
>
> Authors of 13764

---

### Official Review · Reviewer_5N9P · 2025-10-31

**Soundness:** 3
**Presentation:** 2
**Contribution:** 3
**Rating:** 4
**Confidence:** 4

**Summary:**

This paper reassesses the robustness of concept-erased diffusion models and argues that existing concept erasure techniques are fundamentally vulnerable—not because the erasure is ineffective, but because adversarial prompt embeddings capable of restoring erased concepts are widely scattered in the embedding space and largely inherited from the original pretrained model. This reframes the problem: concept erasure does not eliminate latent pathways for restoration, and current evaluations may underestimate attacker capability.

To substantiate this argument, the authors propose RECORD, a white-box, token-level concept restoration attack that replaces the standard projection-based gradient optimization with a tangential coordinate descent strategy. By selecting tokens directly in discrete space through a two-stage ranking-and-evaluation procedure, RECORD avoids projection errors and search-space mismatch that hamper prior methods like P4D and UnlearnDiffAtk.

**Strengths:**

1. The paper is well-written, clearly structured, and easy to follow, making both the motivation and technical contributions accessible to the reader.
2. The work provides an insightful reframing of the vulnerability in concept erasure, highlighting that restoration pathways are largely inherited from pretrained models rather than being caused by the erasure methods themselves.
3. The proposed discrete token-level coordinate descent attack is well-motivated and achieves strong performance, outperforming prior methods and establishing new state-of-the-art results.

**Weaknesses:**

1. The evaluation includes a limited number of erased concepts compared to the experimental settings used in prior baselines. Expanding the set of concepts would strengthen the generality of the conclusions.
2. It would be valuable to include comparisons with more text-encoder-based defense methods. While AdvUnlearn is included, methods such as SAFREE and others are omitted, and adding these would provide a more comprehensive assessment of defense robustness.

**Questions:**

1. Can the proposed method be extended to the text-to-video (T2V) setting? Since the attack operates on the text embedding space, it seems feasible that the approach could transfer to T2V models as well. The authors’ thoughts on this potential extension would be appreciated.

---

> ### Author Response · Authors · 2025-11-21
> **Official Comment by Authors**
>
> We would like to first thank the reviewer for their appreciation of the clarity of our paper (well-written), the core message of our vulnerability studies, and the clear motivation and the strong performance of RECORD.
>
> We would like to address the concerns raised by the reviewer below:
>
> > W1: The evaluation includes a limited number of erased concepts compared to the experimental settings used in prior baselines. Expanding the set of concepts would strengthen the generality of the conclusions.
>
>
> To address the set of the concepts in our experiments, we would like to kindly point out that our evaluation scope (1 style, 3 objects and 1 explicit concept) is largely consistent with other restoration papers, such as P4D, UD, and the restoration papers such as Advunlearn. (the exact concepts used between papers may differ slightly). Below we attach additional experiments on Tench, where RECORD still consistently outperforms other baseline methods. We are willing to add more experiments upon the request from the reviewer.
>
>
> | Tench | Erasediff | ESD  | FMN  | Safree | Salun | Scissorhands | SPM | Stereo |
> | :----- | :---- | :-------- | :--- | :--- | :----- | :---- | :----------- | :-- |
> | **P4D**    | 14.0  | 39.8      | 67.8 | 0.6  | 40.6   | 3.4   | 57.8         | 0.0 |
> | **UD**     | 2.2   | 4.5       | 59.4 | 1.0  | 2.5    | 1.7   | 23.5         | 0.4 |
> | **RECORD** | **64.4**  | **79.0**      | **93.9** | **85.6** | **75.8**   | **7.0**   | **91.2**         | **1.2** |
>
>
> > W2: It would be valuable to include comparisons with more text-encoder-based defense methods. While AdvUnlearn is included, methods such as SAFREE and others are omitted, and adding these would provide a more comprehensive assessment of defense robustness.
>
>
> Regarding additional experiments on text-encoder-based defense methods, we follow the review’s suggestions by presenting the result for SAFREE. We would also like to add that SAFREE is a training-free method, based on subspace projection of the input prompt. We hope this demonstration of RECORD’s generalisability against different forms of concept erasure methods is satisfactory. We also note that RECORD outperforms by up to a factor of **82 times** (garbage truck RECORD vs UD P4D) on SAFREE.
>
>
> | Safree | van Gogh | Nudity | Church | Garbage Truck | Parachute |
> | :-------- | :------------ | :-------- | :-------- | :----------------- | :------------ |
> | **P4D**     | 1.2           | 10.0    | 3.2       | 1.0                 | 2.0            |
> | **UD**       | 2.3           | 9.4      | 1.3       | 1.0                 | 1.3            |
> | **RECORD** | **92.0**	   | **80.2**    | **94.2**     | **82.0**                  | **54.4**          |
>
>
> > Question: Can the proposed method be extended to the text-to-video (T2V) setting? Since the attack operates on the text embedding space, it seems feasible that the approach could transfer to T2V models as well. The authors’ thoughts on this potential extension would be appreciated.
>
>
> Thank you for the insightful question, regarding possible T2V extension of the algorithm, although it is beyond the scope of this paper, it is a natural extension and is something we are actively exploring.
>
> We are delighted to report that we have conducted new additional experiments according to the reviewer’s suggestions. We eagerly await continued feedback, and will follow up with a revised manuscript according to our further discussion with the reviewer.
>
> We hope the experimental results are useful in helping the reviewer reconsider their review and adjust their scores. We are happy to discuss if the reviewer has any further concerns.

---

> > ### Comment · Reviewer_5N9P · 2025-11-27
> >
> > Thank you for the additional experiments; the new results on Tench and SAFREE are helpful and address parts of my concerns.
> >
> > W1: The added concept strengthens your empirical support, but the evaluation scope is still narrower than in several prior works, so my concern about generality remains.
> >
> > W2: Including SAFREE is appreciated; however, I note that the nudity performance reported here differs from the original SAFREE paper, and it is unclear whether this discrepancy arises from differences in implementation, prompt selection, evaluation protocol, or erasure configuration. Clarifying this would be important for a fair comparison.

---

> > > ### Author Response · Authors · 2025-11-28
> > >
> > > We thank the reviewer for their detailed follow up, and are glad that our additional experiments helped address some of their concerns.
> > >
> > > > W1: The added concept strengthens your empirical support, but the evaluation scope is still narrower than in several prior works
> > >
> > >
> > > We would like to thank the reviewer’s acknowledgement of our results on Tench. Although we consider the concepts we investigated in our paper (from art style, objects, to explicit concepts) are representative and general, we thank and will follow the reviewer’s constructive suggestions on the generality. We are currently running additional experiments to further expand the scope (violence, illegal activity, car, french horn). We hope this would convince you regarding the generality of the experiments, and we will update you once we obtain any additional results.
> > >
> > >
> > > > W2: I note that the nudity performance reported here differs from the original SAFREE paper, and it is unclear whether this discrepancy arises
> > >
> > > Unfortunately, we note that the SAFREE paper does not provide the code for running the attacks, nor sufficient details regarding the attacks in their paper for replication. Below we will try to clarify the potential differences in the evaluation protocol with some speculation on the SAFREE’s experimental setting
> > >
> > > There are a few reasons which may have caused discrepancies, the main one is due to the different experiment setup:
> > >
> > >
> > > 1. We observe that e.g. for UnlearnDiffAtk (UD) on ESD, SAFREE paper reports 76.1% in Table 1. This aligns with the number reported in UD's original paper (76.05%). However, UD’s evaluation process may differ quite significantly from ours. UD’s code generates one image to evaluate the prompt every one of its 50 optimization steps, and stops immediately when one image is deemed nude by e.g. a NudeNet classifier. This means each prompt gets evaluated 50 times throughout the optimisation, which generally boosts the ASR. From our experience, it also leads to a significant number of false positives. We note that, if the prompt is only evaluated once at the end of the 50 steps, the ASR can decrease by 3 times (in the case of ESD, to 26.2% vs 76.1% in SAFREE’s paper). A similar technique is also used in P4D’s original codebase, where each prompt generates 3 images, and is deemed unsafe if one of the three images contains the erased unsafe concept. In our experiments, we only evaluate each optimised prompt at the end of the optimisation once by generating 10 images, and reporting the percentage of the unsafe images over the total number of images generated, as stated in **line 419**. This should be the main reason where the discrepancy comes from.
> > >
> > > 2. The followings  are relatively minor differences between the implementations: For any concept erasure methods requiring finetuning, model weights may differ; We use a different set of target prompts from I2P; Because we do not use I2P, we also do not fix the seed when running the experiments; We use a diffusion classifier rather than NudeNet for evaluating the ASR.
> > >
> > > Furthermore, to address any concerns related to the differences between the evaluation processes, we have included **Table 9 in Appendix F** in our initial submission where we tried to evaluate RECORD against P4D and UD under UD’s evaluation pipeline (where each prompt is evaluated 50 times in total) with I2P prompts and NudeNet. We found that under this approach, RECORD can still demonstrate superior performance over other baseline attack methods.
> > >
> > > We hope our response classifies any concern from the reviewer. We are happy to discuss further if the reviewer has other questions.

---

> > > ### Author Response · Authors · 2025-12-03
> > > **Follow-up Comment on W1**
> > >
> > > We would like to follow up with experiments on additional concepts. Due to the time constraint, we present the additional concepts erased on three newest (AdvUnlearn, SAFREE, STEREO) and one classical (ESD) erasure methods.
> > >
> > > | Car | ESD | AdvUnlearn | SAFREE | STEREO |
> > > | :--- | :--- | :--- | :--- | :--- |
> > > | **P4D** | 26.6 | 2.8 | 6.8 | 2.2 |
> > > | **UD** | 8.6 | 0.0 | 2.6 | 2.6 |
> > > | **RECORD** | **53.8** | **60.2** | **63.0** | **4.0** |
> > >
> > > | French horn | ESD | AdvUnlearn | SAFREE | STEREO |
> > > | :--- | :--- | :--- | :--- | :--- |
> > > | **P4D** | 17.4 | 5.6 | 0.8 | 0.4 |
> > > | **UD** | 2.3 | 0.0 | 0.0 | 0.4 |
> > > | **RECORD** | **70.6** | **87.8** | **94.0** | **1.0** |
> > >
> > > | Violence | ESD | AdvUnlearn | SAFREE | STEREO |
> > > | :--- | :--- | :--- | :--- | :--- |
> > > | **P4D** | 28.6 | 7.6 | 25.8 | 7.0 |
> > > | **UD** | 37.8 | 0.0 | 11.8 | 13.2 |
> > > | **RECORD** | **67.2** | **57.2** | **73.0** | **14.6** |
> > >
> > > | Drugs | ESD | AdvUnlearn | SAFREE | STEREO |
> > > | :--- | :--- | :--- | :--- | :--- |
> > > | **P4D** | 45.6 | 12.8 | 8.2 | 5.4 |
> > > | **UD** | 7.6 | 0.0 | 8.6 | 4.0 |
> > > | **RECORD** | **87.6** | **89.0** | **88.0** | **21.2** |
> > >
> > > | Obama | ESD | AdvUnlearn | SAFREE | STEREO |
> > > | :--- | :--- | :--- | :--- | :--- |
> > > | **P4D** | 6.6 | 1.8 | 0.6 | 0.4 |
> > > | **UD** | 2.6 | 0.0 | 1.6 | 0.0 |
> > > | **RECORD** | **30.8** | **15.6** | **98.6** | **3.2** |
> > >
> > > The above results continue to demonstrate the superior performance of RECORD. We hope this can further convince the reviewer about the strong experimental support of our methods.

---

> > > ### Author Response · Authors · 2025-12-03
> > > **Final Comment**
> > >
> > > Dear reviewer 5N9P,
> > >
> > > Unfortunately we are unable to engage with you due to the sudden ICLR rule change. We would like to summarise below
> > >
> > > - We have more than **doubled the scope** of our experiment by providing additional results on **6 concepts** to address your concern, which continues to demonstrate the significantly superior performance of RECORD.
> > > - We have also provided new **SAFREE results on all concepts** evaluated in the paper / rebuttal, which again demonstrates the **significantly superior** performance of RECORD.
> > >
> > > We hope our clarification, together with the new results, can address all of your concerns and further convince you about the contribution of our paper.
> > >
> > > Sincerely,
> > >
> > > Authors of 13764

---

### Official Review · Reviewer_gZBK · 2025-10-31

**Soundness:** 2
**Presentation:** 3
**Contribution:** 2
**Rating:** 2
**Confidence:** 5

**Summary:**

In this paper, the authors explored the reasons of concept-erasure vulnerability and found that even though the adversarial embeddings were far from the original ones, they could also generate the corresponding contents. It inspired the motivation to search adversarial tokens in a boarder semantic space. Therefore, they proposed RECORD, a coordinate descent approach that iteratively optimizes the prompts.

**Strengths:**

1. The paper is well-written, easy for readers to follow.
2. Vulnerability of concept erasure is an important topic. I admire the authors' focus on this field.

**Weaknesses:**

1. The method is somewhat easy. The experiments in Sec 3.1 and 3.2 are intuitive and persuasive, although previous studies have shown this point. However, the proposed method is only borrowed from recent studies, just as the cited papers in Line 296. The authors did not introduce any novel method based on their empirical observation.

2. The evaluation is unfair and insufficient to some extent. The used metric is based on the denoising errors. However, the proposed attacking method is also based on the denoising errors. This has resulted in an optimization process in the evaluation method. Also, the results in Tab 2 reveal that this metric is not accuracy. For nudity, it only achieves 87.6% accuracy. Why not to use the common concept detectors such as NudeNet? It is trained on large data and therefore has excellent detection ability.

**Questions:**

Considering that the authors explored the behaviors of the text embeddings, can we optimize a loss function only on text encoders?

---

> ### Author Response · Authors · 2025-11-21
> **Official Comment by Authors (part 1 / 2)**
>
> We would like to first thank the reviewer for their appreciation of the clarity of the paper, our vulnerability study, as well as the importance of the vulnerability of the concept erasure topic.
>
> We would like to address the concerns raised by the reviewer below:
>
> > W1: The method is somewhat easy. The experiments in Sec 3.1 and 3.2 are intuitive and persuasive, although previous studies have shown this point. However, the proposed method is only borrowed from recent studies, just as the cited papers in Line 296. The authors did not introduce any novel method based on their empirical observation.
>
> Regarding the methodology, we acknowledge RECORD is inspired by optimization techniques on adversarial attacks of language models in **line 151**, but there is no guarantee that similar methods would work for concept restoration in text-to-image diffusion models prior to our work. We also acknowledge that coordinate descent is a classic optimization technique. We would like to politely note that our work is the first that applies the coordinate descent algorithm in the field of concept restoration, and observed significant improvement over the baselines through extensive experimentation and ablation studies. Significant work is involved for transferring such ideas to the modality of text-to-image generation, and we rigorously justified the algorithmic design in Appendix B, C, D, and demonstrated the superior performance of RECORD.
>
>
> Rather than exploring sophisticated new strategies for concept restoration, this paper demonstrates the effectiveness of using classical and simple optimization (i.e. coordinate descent) to address new and challenging questions (i.e. does direct token optimization outperform projection based methods on image generators? Are the results generalizable? Are adversarial prompts transferable to black-box models?).
>
> Furthermore, our work also investigates the root cause of vulnerability in concept-erased models, the first of its kind and a novel contribution of our paper as well. We consider the significant gain of performance over previous SOTA achieved with classic and intuitive optimisation techniques to be a strength of this work.

---

> ### Author Response · Authors · 2025-11-21
> **Official Comment by Authors (part 2 / 2)**
>
> > W2: The evaluation is unfair and insufficient to some extent. The used metric is based on the denoising errors. However, the proposed attacking method is also based on the denoising errors. This has resulted in an optimization process in the evaluation method. Also, the results in Tab 2 reveal that this metric is not accuracy. For nudity, it only achieves 87.6% accuracy. Why not to use the common concept detectors such as NudeNet? It is trained on large data and therefore has excellent detection ability.
>
> Regarding the review’s concern on the classifier used in the evaluation process, we motivate the use of diffusion classifiers in **line 349 - 353**, that we attempt to propose a single, unified evaluation model over a large ensemble of different classifiers from different sources for different concepts, potentially introducing inconsistencies.
>
> Regarding “The used metric is based on the denoising errors. However, the proposed attacking method is also based on the denoising errors”, we respectfully disagree that this leads to any unfair comparison.
>
> First, we would like to politely point out that the loss functions of all restoration methods (P4D, UD, RECORD) are based on denoising errors. In fact, UD’s loss function (Equation 3 in Appendix B) is more similar to the objective of the diffusion classifier (stated in Section 4.1) than that of RECORD or P4D (Equation 4 in Appendix B). We totally understand where the reviewer’s concern is coming from, and if what the reviewer said were true, then our current setup is unfairly advantaging UD over P4D or RECORD. However, as we will demonstrate below, this concern is not true.
>
> Secondly, our ablation study in Appendix B demonstrates UD’s loss function (Equation 3) marginally underperforms when compared with P4D’s / RECORD’s (Equation 4). We therefore evidently believe that this experiment eliminates concerns regarding any unfair evaluation due to the similarity between the objectives of the attacking and the evaluation method, and that the evaluation process does not unfairly advantage or disadvantage any methods in any of our experiments.
>
> Thirdly, regarding the accuracy of the diffusion classifier, we rigorously examined its robustness during the early stage of the project. For NudeNet with a configuration in line with those used in P4D and UD repositories, it achieves an accuracy of **89.8%**, which is very similar to the **87.6%** of the diffusion classifier we used. We would like to further re-assure the reviewer that if there had been any problem regarding the use of the diffusion classifier, we would have (almost trivially) switched back to the traditional classifiers e.g. NudeNet. Instead, we hope this can be counted as one of the contributions of our work.
>
> We are happy to discuss this if the reviewer has any further concerns.
>
> > Question: Considering that the authors explored the behaviors of the text embeddings, can we optimize a loss function only on text encoders?
>
> We sincerely thank the reviewer for carefully reading our paper and providing a very interesting question regarding the feasibility of optimising only on the text encoders, firstly, strategies like “optimising a loss function only on text encoders” are only possible if the erasure method modifies the text encoder (the mapping of the texts to embedding is changed), but not possible if only the denoiser is modified (the mapping remains unchanged). This narrows the question to a simpler objective of searching for the same original text embedding with a concept-erased text encoder, i.e. $\min_{y’} \|\mathcal T(y) - \mathcal T’(y’) \|$, where $\mathcal T$ and $\mathcal T’$ corresponds to the original / concept-erased text encoder, $y$ corresponds to the original erased concept.  We would expect this to underperform, as the optimisation neglects the essential signal from the denoiser on “how the final image might look”, and will only try to find one single embedding $\mathcal T(y)$, which can be very difficult, rather than exploiting other embeddings far-away from $\mathcal T(y)$ that can generate the same erased concept as explained in section 3.2. Indeed our embedding-level attack as shown in the table below confirms this hypothesis, and the difference in performance is rather significant.
>
> | Method | Van Gogh | Church | Garbage Truck | Parachute | Nudity |
> | :-- | :-- | :-- | :- | :-- | :-- |
> | **Text Encoder Only** | 9.8 | 0.0 | 0.0 | 0.6 | 3.4 |
> | **Denoiser + Text Encoder** (Ours, Table 4 in paper) | **99.0** | **100.0** | **92.6** | **44.8** | **96.8** |
>
> We are delighted to report that we have conducted new additional experiments according to the reviewer’s suggestions. We eagerly await continued feedback, and will follow up with a revised manuscript according to our further discussion with the reviewer.
>
> We hope our responses address the reviewer’s concerns and can help them reconsider and adjust their score.  We are happy to discuss if the reviewer has any further concerns.

---

> > ### Comment · Reviewer_gZBK · 2025-11-26
> > **Response to Rebuttal**
> >
> > Thanks very much for your response.
> >
> > 1. I have noticed that many studies have shown that perturbations to the embeddings can allow the model to generate relevant content, and these perturbations are no longer limited to the neighborhood of concept text embeddings. I believe this is due to the black-box nature of the model, which remains difficult to understand and interpret. Your contribution might lie in demonstrating the optimization trajectory, but this is still a phenomenon, and whether it's the root cause of vulnerability remains to be discussed.
> >
> > 2. I did not find the result "89.8%" in the paper. Can you provide more details about it? What images and labels do you use to evaluate the performance of NudeNet? If the data is consistent with the details mentioned in Sec4.1, it brings another question: are ground-truth labels correct? Some prompts cannot lead models to generate corresponding images, for example, some prompts in I2P. Please check it carefully.
> >
> > I did not deny the fact that denoising error can act as a classifier. In fact, this has been proposed in previous research [1]. If you can provide the results using the classifiers, I think it will be more straightforward. Also, for nudity, NudeNet can provide detailed categories, i.e., breast,  but this metric cannot.
> >
> > [1] Your diffusion model is secretly a zero-shot classifier, 2023.
> >
> > I have no other questions. The authors address my question and the concern about the evaluation fairness. Thank the authors for their efforts again.

---

> > > ### Author Response · Authors · 2025-11-26
> > >
> > > We would like to again first thank the reviewer for following up with their valuable comments.
> > >
> > > > 1. I have noticed that many studies have shown that perturbations to the embeddings can allow the model to generate relevant content…Your contribution might lie in demonstrating the optimization trajectory, but this is still a phenomenon, and whether it's the root cause of vulnerability remains to be discussed.
> > >
> > > We would like to respectfully clarify that our contribution regarding the vulnerability studies extends beyond simply demonstrating optimization trajectories. To our best knowledge, our work is the first to exploit perturbations on text embeddings to demonstrate the **pervasiveness** and the **inherited nature** of the vulnerabilities in the embedding space. In **Section 3.2**, we demonstrate vulnerability is pervasive even in regions semantically distant from the target concept. In **Appendix A** we provide empirical evidence that this “vulnerability” is a characteristic inherited directly from the original non-erased model (**line 741 - 743**), which provides an intuitive explanation of the origin of the erasure vulnerability.
> > >
> > >
> > > > 2. I did not find the result "89.8%" in the paper
> > >
> > > Regading 89.8%, we agree that this was not included in the initial submission, but is a new result using NudeNet in response to the reviewer’s concern. This is consistent with 87.6% of the diffusion classifiers, which demonstrates the validity of the use of diffusion classifiers in the evaluation process.
> > >
> > > > for nudity, NudeNet can provide detailed categories, i.e., breast, but this metric cannot.
> > >
> > > Regarding more detailed categories of nudity, we believe it should be possible to engineer the classification classes $Y$ to include more specific body parts, but this is beyond the scope of this work as we focus only on the general concept of nudity.
> > >
> > >
> > > > What images and labels do you use to evaluate the performance of NudeNet? ... are ground-truth labels correct? Some prompts cannot lead models to generate corresponding images, for example, some prompts in I2P
> > >
> > >
> > > We agree with the reviewer that many prompts from I2P do not generate nude images, which is also consistent with our observations. In our work, we use short and very straightforward target prompts for testing the classifiers and to evaluate the ASRs, e.g. “A photorealistic image of a nude woman” for generating nudity images, and e.g. “A photorealistic image of a dressed woman” for generating non-nudity images. Moreover, for testing the classifiers, we generate 500 images, 250 images for each nude / non-nude class. We notice that by using these short and straightforward prompts, the model can consistently generate the corresponding images containing the target concept. In fact, upon careful manual inspection of the generated images, we counted less than 1% (2/250) of the images generated with nudity prompts clearly not containing nudity elements. Therefore when testing the diffusion classifiers, we assume that all images generated with nudity prompts are labelled nude, and vice versa.
> > >
> > >
> > > We would like to further follow the reviewer’s suggestion by providing the ASRs evaluated by **NudeNet** below:
> > >
> > > | Method | ED | ESD | SalUn | SH | SPM | AdvUnlearn | RECE |
> > > | :--- | :---: | :---: | :---: | :---: | :---: | :---: | :---: |
> > > | **P4D** | 0.0 | 22.0 | **1.4** | 2.6 | 44.6 | 18.0 | 12.0 |
> > > | **UD** | 0.0 | 21.0 | 0.8 | 0.4 | 8.2 | 14.6 | 14.2 |
> > > | **RECORD**| 0.0 | **66.6** | **1.4** | **4.0** | **63.4** | **30.6** | **22.4** |
> > >
> > > which also demonstrates the consistently superior performance of RECORD by often a very significant margin.
> > >
> > > We hope our responses address the reviewer’s concerns, and can convince them of the contributions of our work, the insightful vulnerability studies, and the significantly superior performance of the proposed algorithm. We hope the reviewer can adjust their score accordingly, and we are happy to discuss if the reviewer has any further concerns.

---

> > > > ### Comment · Reviewer_gZBK · 2025-11-28
> > > >
> > > > Sorry for the response delay. I have noticed the Openreview bug issue and ICLR has closed the review editing button. I will release my further comment once reciving further notifications from PCs. Thank the authors again for their hard efforts.

---

> > > > > ### Author Response · Authors · 2025-12-03
> > > > > **Final Comment**
> > > > >
> > > > > Dear reviewer gZBK,
> > > > >
> > > > > Unfortunately we are unable to engage with you due to the sudden ICLR rule change. We would like to summarise below
> > > > >
> > > > > - We believe our clarifications have addressed your concerns regarding the novelty of RECORD, the first coordinate-descent algorithm for concept restoration. We are also confident that our vulnerability study, which demonstrates its **pervasiveness** and the further insights into its **origin** is the first of its kind in this field.
> > > > > - We have also provided new results regarding NudeNet, which demonstrates the competitive performance of our diffusion classifier, and continues to demonstrate the significantly superior performance of RECORD. We thank you for acknowledging that your initial concerns regarding evaluation fairness has been addressed.
> > > > > - We have provided additional experiments testing the text-encoder-loss-only approach, which did not work. We thank you for acknowledging that this question has also been addressed.
> > > > >
> > > > > We hope our clarification, together with the new results, can address all of your concerns and further convince you about the contribution of our paper.
> > > > >
> > > > > Sincerely,
> > > > >
> > > > > Authors of 13764

---

### Meta-Review · Area_Chair_AXt5 · 2026-01-02

**Summary:**

The reviewers’ concerns mainly focus on limited methodological novelty and technical insight. The proposed approach builds on existing embedding perturbation and adversarial optimization techniques without introducing a fundamentally new method. Reviewers also note that the experimental scope is narrower than in some prior work, raising questions about generality. Reviewers remain unconvinced that the analysis establishes the root cause of vulnerabilities in concept-erased models.

**Reviewer Concerns:**

The core optimization strategy is adapted from existing embedding perturbation and adversarial prompt optimization techniques. In addition, although the paper provides extensive empirical evidence on the pervasiveness and inheritance of vulnerabilities in concept-erased models, the claim of identifying the root cause of these vulnerabilities remains partially unresolved. The current analysis characterizes the phenomenon rather than offering a definitive causal or theoretical explanation.

**Reviewer Scores:**

The additional experiments and clarifications would partially address concerns about evaluation fairness and empirical validity, but the central issues regarding methodological novelty, technical insight, and claims about root causes would remain. Reviewers might have adjusted their scores slightly upward, but most likely only to the level of a borderline reject, rather than to a clear accept.

---

### Decision · Program_Chairs · 2026-01-26

Reject